# Autoantibodies to IL-1Ra and PGRN in severe COVID-19 are associated with inflammation-induced hyperphosphorylated antigen isoforms

SARS-CoV-2 infection affects multiple immune mechanisms and leads to severe COVID-19 and death, in part related to infection-induced or pre-existing autoantibodies. Here, we describe severe COVID-19 to associate with auto-antibodies against interleukin-1 receptor antagonist (IL-1Ra) and progranulin (PGRN), endogenous antagonists of IL-1 and TNF signaling, respectively. These autoantibodies coincide with hyperphosphorylation of IL-1Ra (Thr111) or PGRN (Ser81), form immune complexes independent of phosphorylation, reduce antigen plasma levels, and permit enhanced IL-1 and TNF signaling. Using phage-display selected Fabs specific for hyperphosphorylated isoforms, we track phospho-antigens and autoantibodies in a German national pandemic network cohort. Most seropositive patients show both autoantibodies. Levels peak at baseline and decline over 12 months, with phospho-antigens decreasing before autoantibodies. Seropositivity associates with hyperinflammation and cytokine profiles. Importantly, signaling by key inflammatory cytokines induce IL-1Ra and PGRN hyperphosphorylation in healthy monocytes, but require up to 1000-fold higher doses than in monocytes from previously seropositive severe COVID-19 survivors.

Infection with SARS-CoV-2 can result in a wide spectrum of clinical manifestations (COVID-19) and severity. Those can range from a completely asymptomatic course, to mild cold symptoms or attenuation of the sense of taste and smell, to acute respiratory distress syndrome (ARDS), the latter often associated with thromboembolic complications[1,2]. Patients requiring intensive care treatment often present with a hyperinflammatory condition, with some immune-phenotypic overlap with infection-associated secondary hemophagocytic lymphohistiocytosis (ia-HLH)[3–5].

Already early into the pandemic, SARS-CoV-2 infection and particularly severe and fatal courses of COVID-19 have been associated with several autoantibody phenotypes. Exuberant B-cell responses were regularly found in COVID-19[6], along with a high prevalence of antiphospholipid[7], anti-Ro/La-[8] or anti-annexin-V-antibodies[9]. Plasma screening of barcoded exoproteins displayed on yeasts identified several autoantibodies directed against immune- and tissue-related proteins[10]. A major finding was the discovery of autoantibodies against type I interferon in patients with severe courses of COVID-19[11], proposing impaired type I interferon responses as a key factor for an inefficient immune response[12,13], with an autoantibody phenotype in a subpopulation of patients[11]. Importantly, these autoantibodies were already pre-existing in patients and pre-disposed those individuals for more severe courses of viral infection, likely due to attenuated viral clearance [14–17].

In the context of severe (hyper)inflammatory courses of SARS-CoV-2 infection in children (multisystem inflammatory syndrome in children, MIS-C), mRNA-vaccination-associated myocarditis, as well as an inflammatory condition termed Still's disease, we already

✉e-mail: lorenz.thurner@uks.eu

investigated and reported proinflammatory antibodies targeting the interleukin 1 receptor antagonist (IL-1Ra)[18–20]. These antibodies immune-complexed and depleted peripheral IL-1Ra, thereby impairing its bioactivity, facilitating unrestricted IL-1β signaling[18–20]. Outside the context as described above, autoantibodies neutralizing IL-1Ra have only been described in IgG4-related disease[21]. Importantly, those as well as anti-IL-1Ra antibodies identified by us appear to share epitope specificities[18,19,21].

Similar to IL-1Ra, progranulin (PGRN, also called proepithelin) can antagonize TNF and TNF-like cytokine 1 A (TL1-A) signaling via direct binding to TNFR1, TNFR2 and DR3[22,23]. Clinical relevance of PGRN as modulator of TNF-signaling has been demonstrated in several inflammatory mouse models, including LPS-induced lung injury mimicking ARDS[24–27]. On the molecular level, both PGRN and TNF bind to cysteine-rich domain 2 and 3 of TNFR[28,29]. Importantly, we previously identified neutralizing autoantibodies directed against progranulin (PGRN) in sera from patients with primary small vessel vasculitis as well as other rheumatic and autoimmune conditions[30–34]. Mechanistically, we already linked the proinflammatory effect of neutralizing anti-PGRN-antibodies with TNF-induced cytotoxic effects on WEHI-S cells as a reporter for detrimental TNF signaling and downmodulation of FoxP3 in CD4⁺CD25[hi] Tregs[32].

With hyperinflammatory states often observed in patients with SARS-CoV-2 infection, and in light of similarities between this viral condition, vasculitis, autoimmune and autoinflammatory diseases, in the present study, we set out to investigate the occurrence of antibodies directed against previously described anti-inflammatory antigens, including progranulin and IL-1Ra, in the context of COVID-19. We find severe COVID-19 to be associated with autoantibodies targeting IL-1Ra and PGRN, which coincide with hyperphosphorylated antigen isoforms and reduce circulating antagonist levels through immune complex formation, thereby enhancing IL-1β and TNF signaling. Phospho-antigens and autoantibodies peak early in disease and decline over time, with changes in antigens preceding changes in antibody titers. Seropositivity for IL-1Ra- and PGRN-targeting autoantibodies associates with distinct inflammatory mediator profiles over time, while monocytes from previously affected individuals display a markedly increased sensitivity to cytokine-induced hyperphosphorylation compared with healthy controls. Together, these findings indicate a transient, disease-imprinted dysregulation of inflammatory control mechanisms. They suggest that inflammation-driven post-translational antigen modification and autoimmunity can contribute to prolonged immune imbalance and disease severity.

## Results

### Anti-PGRN and anti-IL-1Ra antibodies in severe COVID-19 frequently coincide with atypical autoantigen isoforms

For the purpose of this study we compiled a discovery cohort comprising 30 COVID-19 patients hospitalized in intensive care unit (ICU) as well as 28 ICU control patients outside of a COVID-19 context (Supplementary Data 1 and 2; Supplementary Fig. 1). Using a detection system based on a previously reported in-house ELISA[34], PGRN-reactive antibodies were detected in plasma of 11 out of 30 discovery cohort patients (36.7%; Fig. 1A). In all patients' plasma seropositive for PGRN-reactive antibodies we also observed antibodies binding to IL-1Ra. Three patients' plasma presented with only IL-1Ra-reactive antibody (total number with anti-IL-1Ra IgG: 14 out of 30 patients, 46.7%) (Fig. 1A). Titers of anti-PGRN- and anti-IL-1Ra antibodies ranged from 1:1600 up to 1:3200 and from 1:800 to 1:1600, respectively, and in the vast majority cases belonged to IgM and several IgG subclasses (Supplementary Fig. 2).

Native gradient western blots of discovery cohort patients' plasma revealed reduced levels of free (non-immune complexed) granulin (GRN) and IL-1Ra, coinciding with IgG and IgM immune complexed antigen, when plasma samples tested positive for anti-PGRN and anti-

IL-1Ra antibodies (Fig. 1B and Fig. 1D). We did not observe such in seronegative patients' plasma samples (Fig. 1B and Fig. 1D).

Our laboratory has extensive experience in identifying post-translational modification-related protein isoforms and linking those with autoantibody pathology[35–38]. To test for potential atypic PGRN and IL-1Ra isoforms in autoantibody-positive COVID-19 patients as previously also observed by us in SARS-CoV-2-related and unrelated conditions[18–20,29], we applied isoelectric focusing (IEF) of discovery cohort plasma samples. This analysis demonstrated a more negatively charged PGRN isoform exclusively in anti-PGRN-antibody-positive plasma samples (Fig. 1C). Similarly, we observed a third and more negatively charged IL-1Ra isoform only in patients' plasma seropositive for anti- IL-1Ra antibodies (Fig. 1E).

For all tested discovery cohort samples, seropositivity as determined by in-house ELISA (Fig. 1A), biochemical detection of IgM/IgG-antigen immune complexes (Fig. 1B and Fig. 1D) and occurrence of atypical protein isoforms (Fig. 1C and Fig. 1E) largely converged. As one exception to this, patient 4 tested negative for anti-IL-1Ra antibodies by ELISA (Fig. 1A) but revealed IL-1Ra-IgM/IgG immune complexes (Fig. 1D) as well as an atypical IL-1Ra isoform in plasma (Fig. 1E).

### Validation cohort analysis confirms autoantibody prevalence, specificity and function as well as coincidence with atypical autoantigen isoforms in severe COVID-19

Next to the discovery cohort, prevalence of anti-PGRN and anti-IL-1Ra antibodies according to COVID-19 severity was analyzed in two independent validation cohorts (Supplementary Fig. 1 and Fig. 1F). In both validation cohorts, anti-PGRN and anti-IL-1Ra antibodies were most frequently detected in plasma of patients hospitalized with severe course of COVID-19 (validation cohort 1 ($n = 64$: anti-PGRN: 25/64 (39%); anti-IL-1Ra: 32/64 (50%); Supplementary Data 3; validation cohort 2 ($n = 120$), anti-PGRN: 72/120 (60%); anti-IL-1Ra: 76/120 (62.5%), Supplementary Data 4; Supplementary Fig. 1 and Fig. 1F). Patients with moderate (validation cohort 1 ($n = 126$;, anti-PGRN 16/126 (12.7%); anti-IL-1Ra: 5/126 (3.9%), Supplementary Data 3; validation cohort 2 ($n = 79$); PGRN antibodies: 1/79 (1.3%); IL-1Ra antibodies: 10/79 (12.7%), Supplementary Data 4), or mild (validation cohort 1 ($n = 105$): anti-PGRN: 4/105 (3.8%); anti-IL-1Ra: 2/105 (1.9%), Supplementary Data 3; validation cohort 2 ($n = 79$); PGRN antibodies: 4/79 (5.1%); IL-1Ra antibodies: 0/79 (0%), Supplementary Data 4) COVID-19 disease course revealed markedly lower anti-PGRN or anti-IL-1Ra frequencies (Supplementary Fig. 1 and Fig. 1F).

In contrast, no anti-PGRN and anti-IL-1Ra antibodies were detected in several control cohorts, including healthy pre-pandemic controls ($n = 188$), healthcare workers following 2 vaccinations against SARS-CoV-2 ($n = 40$), and SARS-CoV-2-exposed children and adolescents enrolled in the COKIBA registry ($n = 146$). Anti-PGRN and anti-IL-1Ra antibodies were detected infrequently in patients with a pulmonary focus included in two prepandemic multi-center sepsis trials (SISPCT; NCT00832039 ($n = 121$): anti-PGRN: 3/121 (2.5%); anti-IL-1Ra: 2/121 (1.7%); and MaxSep; NCT00534287 ($n = 53$): anti-PGRN: 1/53 (1.9%); anti-IL-1Ra: 1/53 (1.9%)) (Supplementary Fig. 1 and Fig. 1F).

Throughout and similar to observations linked with the discovery cohort, titers of anti-PGRN- and anti-IL-1Ra antibodies ranged from 1:1600 up to 1:3200 and from 1:800 to 1:1600, respectively, and in the vast majority of cases belonged to IgM and several IgG subclasses (Supplementary Fig. 3).

Severe COVID-19 patients' plasma included in validation cohort 1 was also screened for reactivity to a number of possible antigens with reported anti-inflammatory roles (CD40, IL-10, IL-18bp, IL-22bp, IL-36Ra, and serpin-B1) or relevance to COVID-19 (IFNα2, IFNω, IFNγ). In this analysis, two out of 46 patients revealed IFNα2-binding antibodies (Supplementary Fig. 4A-H).

Further, to exclude a potential artifact or epiphenomenon, cross-reactivity of PGRN- or IL-1Ra-autoantibodies with structural proteins of

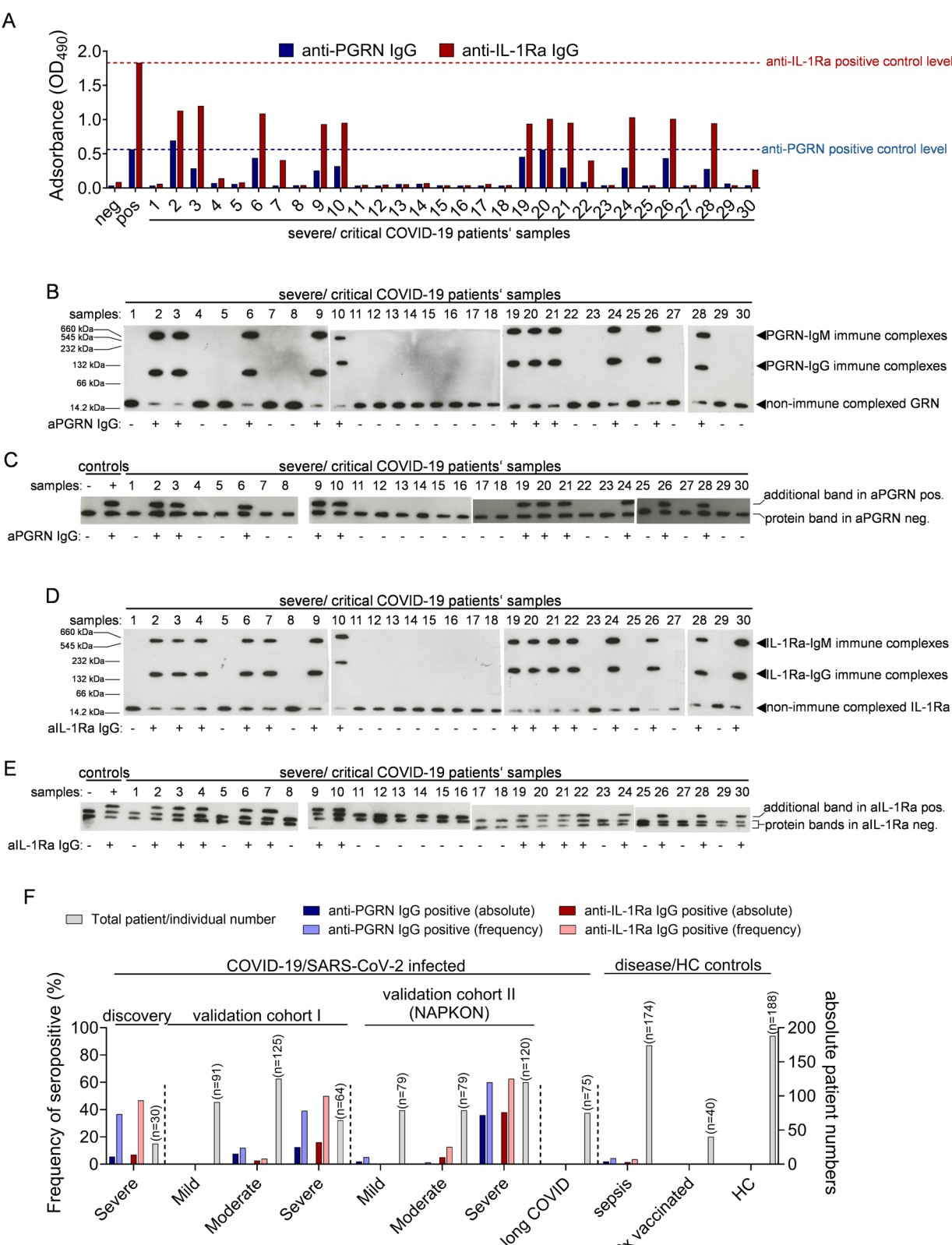

SARS-CoV-2 was ruled out. Anti-PGRN and anti-IL-1Ra IgG purified from patients' plasma did not cross-react with recombinant His-tagged SARS-CoV-2 S1-, S2-, E- or M-proteins (Supplementary Fig. 5A, B). In addition, antibodies against SARS-CoV-2 S1-, S2-, or M-proteins as well as against human IL-1Ra could not be adsorbed by immobilized PGRN or IL-1Ra, but were detectable in the eluate of samples from patients with severe or critical COVID-19, excluding cross-reactivity of anti-PGRN and anti-IL-1Ra (Supplementary Fig. 5A, B). To further confirm the specificity of the identified autoantibodies, we performed competition experiments for binding to plate-bound antigen of IgG purified from patients' plasma *vs.* commercial anti-IL-1Ra (Supplementary Fig. 5C, D) or anti-PGRN mAbs (Supplementary Fig. 5E, F). These

**Fig. 1 | Anti-PGRN and anti-IL-1Ra antibodies in severe COVID-19 frequently coincide with atypical autoantigen isoforms. A** Anti-PGRN and anti-IL-1Ra IgG levels in plasma samples of patients with severe and critical COVID-19 (*n* = 30) enrolled in discovery studies were assessed by in-house ELISA. Anti-PGRN and anti-IL-1Ra positive control levels are indicated as dashed lines. **B,D** Next to ELISA, severe and critical COVID-19 discovery cohort plasma samples (*n* = 30) were also assessed for presence of (**B**) anti-PGRN and (**D**) anti-IL-1Ra IgM and IgG immune complexes using non-reducing native PAGE followed by western blot. **B** Patients' plasma can contain both processed granulins (GRN, also termed epithelin, approx. 10 kDa) as well as progranulin (PGRN, approx. 80 kDa), see Fig.S8A. **C,E** Further, severe and critical COVID-19 discovery cohort plasma samples (*n* = 30) were tested by IEF for occurrence of (**C**) PGRN and (**E**) IL-1Ra isoforms. **F** Analysis of two independent validation cohorts as well as disease and healthy controls (HC) for presence of anti-PGRN and anti-IL-1Ra antibodies as determined by in-house ELISA. Both frequencies (%) of seropositive among all tested samples in each group (left y-axis) as well as absolute numbers of seropositive individuals (right y-axis) are indicated.

experiments demonstrated that purified IgG could outcompete commercial mAb binding in a concentration-dependent manner and vice versa (Supplementary Fig. 5C–F).

On a qualitative level, the reduction of free (non-immune complexed) PGRN and IL-1Ra in autoantibody-seropositive patients' plasma included in the discovery cohort was already visible in western blots of non-reducing native gradient gels (Fig. 1B, D). Similarly, purified antigen (PGRN and IL-1Ra) from validation cohort patient's plasma was immune complexed by plasma IgG and IgM (Supplementary Fig. 6A) and IL-1Ra immune complexed with IgM and IgG could be detected in seropositive patient's plasma (Supplementary Fig. 6B).

To further characterize the observed immune complexes in patients' plasma, we performed size-exclusion chromatography in order to separate non-antibody-bound and immune-complexed antigens from anti-IL-1Ra/anti-PGRN positive (Supplementary Fig. 7B, C) as well as seronegative patients (Supplementary Fig. 7D, E). In case of immune complexed IL-1Ra, we observed three major peaks eluting from seropositive plasma, corresponding to non-immune complexed protein ( ~ 20 kDa), IgG- ( ~ 177 kDa), and IgM-complexed ( ~ 870 kDa) IL-1Ra (Supplementary Fig. 7B), which we validated by immunoblotting (Supplementary Fig. 7F, left panel). Similarly, PGRN eluted as unbound protein ( ~ 15 kDa: granulin), IgG- ( ~ 202 kDa), and IgM-complexed ( ~ 903 kDa) PGRN (Supplementary Fig. 7C, F, right panel). From seronegative plasma proteins eluted as single peaks (Supplementary Fig. 7D, E) corresponding to IL-1Ra (approx. 18 kDa) and unprocessed PGRN (approx., 80 kDa; Supplementary Fig. 7G). Protein fractions as identified by gel filtration corresponded to protein bands as identified in the analysis of COVID-19 patients' plasma in Fig. 1B, D (Supplementary Fig. 7F, H).

Furthermore, in order to verify respective sizes of immune complexes detected in patients' plasma, we generated artificial immune complexes using recombinant antigen and mono- or poly-clonal commercial IL-1Ra or PGRN-specific immunoglobulins (Supplementary Fig. 8). These experiments confirmed the banding patterns of Ig-associated species as observed with immune complexes detected in patients' plasma according to native gradient PAGE (Supplementary Fig 8A, B, D, E) or native PAGE at fixed resolution and subsequent western blotting (Supplementary Fig. 8C). It is of note, though, that in seronegative COVID-19 patients included in the initial analysis (Fig. 1B) we only observed a band resembling non-immune complexed granulin (GRN, processed PGRN), whereas in subsequently conducted gel filtration experiments (Supplementary Fig. 7E, G) and analysis of artificial immune complexes (Supplementary Fig. 8A) we also observed protein bands resembling both non-immune-complexed GRN and PGRN.

When analyzed by commercial ELISAs PGRN levels were decreased by more than 90% in the antibody-positive patients (mean: 21.8 ng/mL), compared to (i) plasma of anti-PGRN-antibody-negative patients (mean: 245.8 ng/mL) and (ii) plasma of seronegative patients hospitalized in ICU outside COVID-19 context (mean 212.2 ng/mL) (Supplementary Fig. 9A). Similarly, IL-1Ra plasma levels were decreased by 78% in anti-IL-1Ra-antibody-positive patients (mean: 258.0 pg/ml), compared to plasma of seronegative patients (mean 1933 pg/ml) (Supplementary Fig. 9E). Anti-PGRN and anti-IL-1Ra antibodies in patients' plasma also resulted in a functional impairment of PGRN and IL-1Ra bioactivity, as determined by TNF- and IL-1-signaling reporter cell lines, respectively (Supplementary Fig. 9B–D, F, G). In these assays, clinically approved drugs, such as Anakinra (recombinant human IL-1Ra), Adalimumab (anti-TNF mAb) or Etanercept (TNFR1-IgG Fc fusion protein) could reverse the impact of anti-PGRN (Supplementary Fig. 9C, D) or anti-IL-1Ra autoantibodies (Supplementary Fig. 9F, G) on IL-1β- or TNF-signaling, respectively.

Importantly, we observed robust quantification of variation in PGRN and IL-1Ra plasma levels to markedly depend on the applied methodology. Compared to several commercial ELISAs, bead array assays failed to demonstrate a depletion of plasma antigen in seropositive samples (Supplementary Fig. 10A, E). Beyond, these assays proofed as rather insensitive in spike-recovery of serial dilutions of commercial recombinant PGRN or IL-1Ra, compared to ELISA (Supplementary Fig. 10B–D, F–H). Further, upon adding commercial anti-PGRN or anti-IL-1Ra antibodies or antibodies purified from plasma to samples spiked with recombinant PGRN or IL-1Ra, this did not result in an antigen depletion visible in bead array assay, but in ELISA (Supplementary Fig. 10B, C, F, G).

## IL-1Ra isoforms result from phosphorylation in serine 97 and threonine 111

Corresponding to observations with PGRN in IEF of patients' plasma (Fig. 1C), anti-IL-1Ra seropositive samples revealed an additional, more negatively charged protein band for a third IL-1Ra isoform (Fig. 1E). To understand whether this protein isoform is also due to additional phosphorylation as previously reported in context with PGRN[29], we pretreated patients' plasma samples with alkaline phosphatase prior to IEF. This resulted in the disappearance of both the second and third protein bands, resembling canonical phosphorylation as well as hyperphosphorylation of IL-1Ra, respectively (Fig. 2A). To identify the site of hyperphosphorylation, isolated monocytes of one patient with anti-IL-1Ra antibodies and one healthy control were transfected with several C-terminally FLAG-tagged constructs encoding IL-1Ra, including point-mutations resulting in an alanine scan of potential phosphorylation sites. IEF of cell lysates using anti-FLAG as primary antibody confirmed serine in position 97 as canonical, and threonine in position 111 as the additional site of phosphorylation (hyperphosphorylation) (Fig. 2B, upper panel). Strikingly, hyperphosphorylation of IL-1Ra only occurred when the protein was expressed from monocytes isolated from COVID-19 patients, but not healthy controls (Fig. 2B, lower panel).

To further study anti-IL-1Ra responses and hyperphosphorylation of antigen also in larger cohorts, we selected T111[phos]-specific Fabs by phage display screening of a human Fab-library (Fig. 2C). Selected Fab-expressing phage clones were tested on synthetic peptides with or without phosphorylation in T111, and specificity for full-length hyper-phosphorylated IL-1Ra was confirmed by IEF (Fig. 2D). Testing discovery cohort samples using the selected phospho-specific Fabs for T111[phos] IL-1Ra as well as previously described Fabs for S81[phos] PGRN[29] and aligning these data with anti-PGRN and anti-IL-1Ra antibody data further validated Fab-specificity (Fig. 2E).

When using wt-IL-1Ra or T111[phos] IL-1Ra-specific Fabs in competition experiments with ex vivo IgG, we observed antibodies in patients' plasma to compete with both wt-IL-1Ra- as well as T111[phos] IL-1Ra-

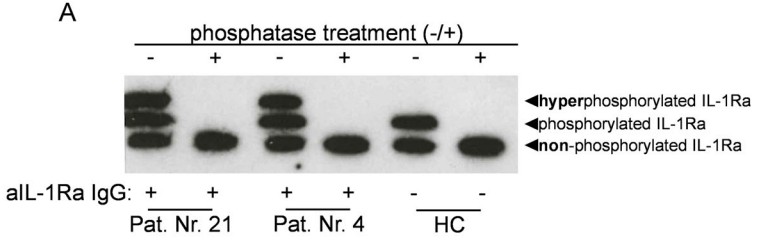

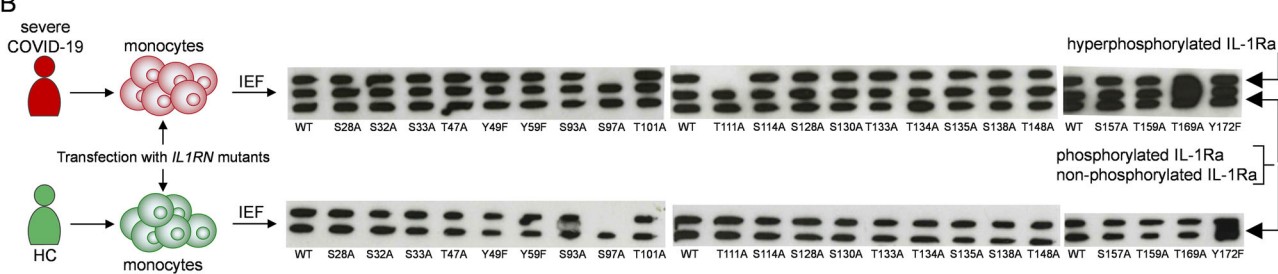

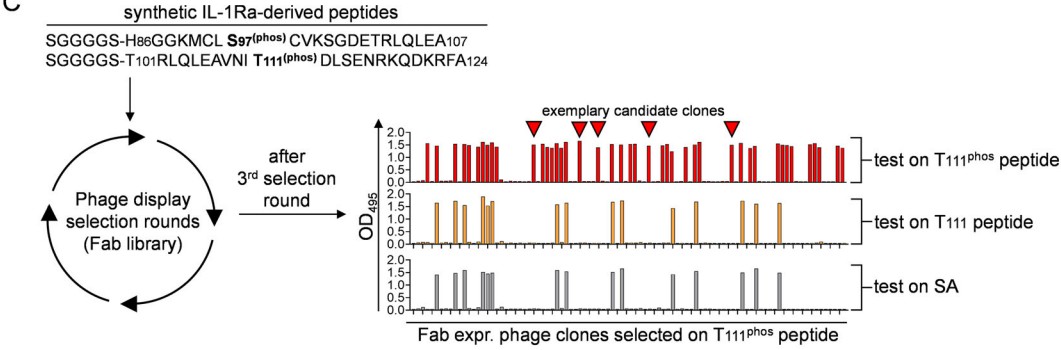

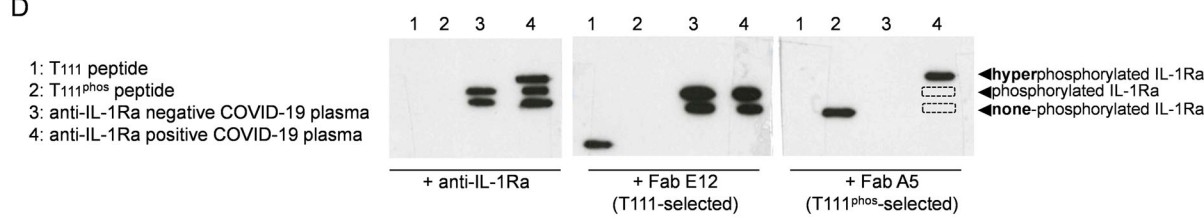

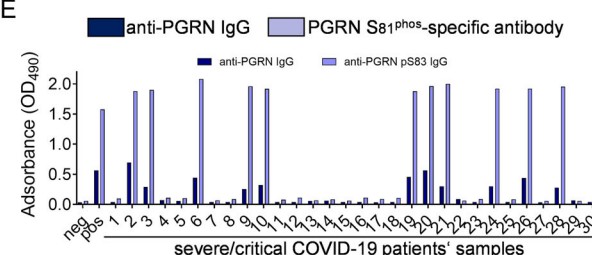

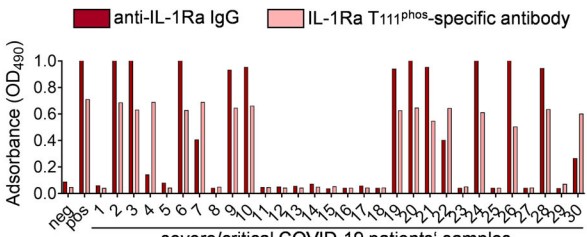

**Fig. 2 | IL-1Ra isoforms result from phosphorylation in serine 97 and threonine 111. A** IEF of phosphatase-treated or untouched selected anti-IL-1Ra seropositive (ELISA) plasma samples (discovery cohort patients). **B** Primary monocytes from severe COVID-19 patients' or HC monocytes and transfected with mutants of IL-1Ra, with all possible phosphorylation sites exchanged for an alanine residue. Protein in cell lysates was tested by IEF for IL-1Ra isoforms. **C** Phage-display selection of IL-1Ra T111$^{phos}$-specific Fabs. **D** Exemplary western blot of IEF using Fabs with reactivities for non-T111$^{phos}$ (E12) or T111$^{phos}$ (A5) for staining. **E** Functionality of selected phospho-antigen specific Fabs in an ELISA format was tested using discovery cohort severe and critical COVID-19 patients' plasma samples ($n = 30$). Data show recognition of recombinant PGRN or IL-1Ra by IgG in patients' plasma (dark blue or dark red bars) side-by-side with the presence of hyperphosphorylated PGRN or IL-1Ra in those plasma samples (light blue/light red bars).

specific Fabs for binding to plate-bound canonically phosphorylated or hyperphosphorylated IL-1Ra, respectively (Supplementary Fig. S11).

In the IEF analysis of validation cohort samples, we similarly observed phosphorylation-related additional protein bands with PGRN and IL-1Ra in seropositive individuals (Supplementary Fig. 12A, B). Upon culturing primary monocytes from HC and COVID-19 patients with panels of kinase and phosphatase inhibitors followed by IEF, we identified phospho-kinase (PK) A to drive canonical IL-1Ra phosphorylation in serine 97 (Supplementary Fig. 12C). Additional phosphorylation in threonine 111 was mediated by PKC, instead. Both sites were de-phosphorylated by protein phosphatase 2A (Supplementary Fig. 12C). Further, T111$^{phos}$-specific Fabs immunoprecipitated IL-1Ra in cell lysate together with either subunits of PP2A or PKC. This was not observed when performing IP-experiments with non-phosphorylation-specific Fabs (T111; Supplementary Fig. 12D).

### Decline of hyperphosphorylated antigen precedes loss of seropositivity

Following discovery cohort analysis, presence of anti-PGRN and anti-IL-1Ra antibodies in particularly severe courses of COVID-19 compared to (severe) other infectious and healthy controls has been validated in validation cohort 1 (Fig. 1F). Analysis of a second, independent validation cohort (cohort 2) confirmed these findings (Fig. 1F). On top, this cohort which originates from the German national pandemic network (NAPKON) offered the possibility for longer term follow-up by access to three- and 12-month biosamples as well as multiple paired clinical and laboratory data. In total, baseline samples of 280 patients with COVID-19 (severe: $n = 120$; moderate: $n = 79$; mild: $n = 79$), three-month ($n = 241$), and 12-month follow-up ($n = 183$) samples were analyzed (Fig. 3A). Using phospho-specific anti-PGRN S81$^{phos}$ or anti-IL-1Ra T111$^{phos}$ Fabs, we tracked PGRN or IL-1Ra hyperphosphorylated isoforms over time in NAPKON plasma samples, with side-by-side analysis of total PGRN or IL-1Ra plasma levels (Fig. 3B, C). In anti-PGRN- and anti-IL-1Ra antibody-positive patients' plasma ($n = 74$ and $n = 90$) phospho-antigen specific Fabs (anti-PGRN S81$^{phos}$ and anti-IL-1Ra T111$^{phos}$) revealed highest reactivities with antigen in patients' plasma at COVID-19 baseline (Fig. 3B, C left panels). In comparison, Fab-binding to respective plasma antigen was significantly lower at three- and 12-month follow-up (Fig. 3B, C left panels). Anti-IL-1Ra T111$^{phos}$ Fab-reactivities with plasma antigen at 12-month follow-up was still significantly elevated over those observed with seronegative individuals, which was not observed with anti-PGRN S81$^{phos}$ Fab-binding to PGRN in patients' plasma at one-year follow-up visit (Fig. 3B, C left panels).

Vice versa, in anti-PGRN- and anti-IL-1Ra antibody-positive patients' plasma (n = 74 and n = 90), we observed the lowest PGRN (median: 36 ng/mL) and IL-1Ra plasma levels (median: 141 pg/mL) at COVID-19 baseline (Fig. 3B, C right panels). These levels recovered already at the three-month follow-up visit (PGRN: 130 ng/mL; IL-1Ra: 1051 pg/mL) and remained in range with seronegative patients' baseline plasma levels (PGRN: 131 ng/mL; IL-1Ra: 1362 pg/mL) also at one-year follow-up (PGRN: 159 ng/mL; IL-1Ra: 1522 pg/mL) (Fig. 3B, C right panels).

Individual follow-up data for anti-PGRN and anti-IL-1Ra antibody titers (Fig. 3D, E left panels), anti-PGRN S81$^{phos}$ and anti-IL-1Ra T111$^{phos}$ reactivities in patients' plasma (Fig. 3D, E center panels), and PGRN or IL-1Ra plasma levels (Fig. 3D, E right panels) phenocopy the observed cohort effects (Fig. 3B, C).

At baseline, most patients revealed anti-PGRN and anti-IL-1Ra-antibodies of the IgM and IgG class (anti-PGRN: $n = 65$ of 74; anti-IL-1Ra: $n = 78$ of 90), and a few also of the IgA class (anti-PGRN: n = 7 of 74; anti-IL-1Ra: $n = 9$ of 90) (Fig. 3F, G). At three-month follow up IgM and IgA were less frequently detected and a number of patients already presented with anti-PGRN and anti-IL-1Ra antibodies of the IgG class only (anti-PGRN: $n = 22$ of 55; anti-IL-1Ra: $n = 30$ of 73) (Fig. 3F, G). At 12-month follow-up, only anti-PGRN- and anti-IL-1Ra antibodies of the IgG

class were detected (anti-PGRN: $n = 31$ of 31; anti-IL-1Ra: $n = 42$ of 42) (Fig. 3F, G).

### Autoantibody seropositivity and presence of hyperphosphorylated antigen associates with (hyper)inflammatory markers and specific cytokine signatures

Baseline and follow-up samples of the NAPKON validation cohort were analyzed for association with markers of (hyper)inflammation (CRP, IL-6, ferritin, PCT and hemoglobin levels), according to seropositivity for anti-PGRN and/or anti-IL-1Ra antibodies or none of both (Fig. 4A and Supplementary Fig. 13). Particularly in double seropositive patients we observed these inflammatory markers to be significantly altered compared to seronegative samples (elevated: CRP, IL-6, ferritin, PCT; decreased: hemoglobin) or elevated by trend over samples which tested positive for anti-IL-1Ra antibodies exclusively (IL-6, $p = 0.067$) (Fig. 4A).

Further, we associated cyto- and chemokine levels determined by multiplexed bead array assay in NAPKON samples with autoantibody-status (Fig. 4B and Supplementary Fig. 13). Comparable to observations on clinical routine markers of inflammation, we found selected cyto- and chemokine levels to be significantly altered in double seropositive patients' samples (elevated: IL-6, IL-18, IL-27, CXCL9; decreased: CCL22) or elevated by trend (IL-7, $p = 0.061$) (Fig. 4B). In follow-up samples of seropositive individuals, changes in inflammatory markers as well as cyto- and chemokine levels observed at baseline normalized to levels observed in seronegative patients (Supplementary Fig. 13A–C).

Global correlation analysis of cyto-and chemokine levels in double seropositive NAPKON patients' samples at baseline revealed multiple associations (Fig. 4C), of which many were still present in follow-up samples, albeit with overall weakened correlation (Supplementary Fig. 13D). When linking presence of hyperphosphorylated IL-1Ra as determined by T111$^{phos}$-specific Fabs with cyto- and chemokine levels in respective samples, we found presence of a hyperphosphorylated IL-1Ra isoform to significantly associate with several inflammatory mediators (IL-13, MIP1b, VEGFA, IL-4 and IL-17A) (Fig. 4D, E, upper panels). Of those, only IL-13 was also associated with the parallel decrease in IL-1Ra plasma levels as determined by commercial ELISA (Fig. 4D, E, lower panels). In follow-up samples, comparable significant associations with both the presence of hyperphosphorylated IL-1Ra as well as IL-1Ra plasma level decrease were seen with FLT3L and IL-8, but not IL-13 (Supplementary Fig. 13D, F). Of note, such associations with FLT3L where already seen in baseline samples, albeit below the significance level (Fig. 4D, E).

In turn, when linking presence of hyperphosphorylated PGRN (determined by S81$^{phos}$-specific Fabs) with cyto- and chemokine levels in respective samples, we found presence of the hyperphosphorylated PGRN isoform to associate best with MCP-1/CCL2 by trend (Fig. 4F, G, left panels), and to significantly associate with a paralleling decrease in PGRN plasma levels as determined by commercial ELISA (Fig. 4F, G, right panels). In follow-up samples, we observed no comparable significant associations with both the presence of hyperphosphorylated PGRN and the decrease in PGRN plasma level (Supplementary Fig. 13G, H).

In addition to peripheral blood proteomics, we also analyzed whole-blood gene expression by RNA sequencing. In the NAPKON cohort, positive autoantibody status associated with a pronounced upregulation of the top 50 most significantly differentially expressed genes (Supplementary Fig 14A, B). Gene set enrichment analysis revealed a prominent association of severe disease with platelet activation and blood coagulation pathways (Supplementary Fig. 14C). However, when comparing severely ill COVID-19 patients seropositive for autoantibodies with equally sick COVID-19 patients but seronegative for anti-PGRN/anti-IL-1Ra, we observed no significant DEGs (Supplementary Fig. 14D).

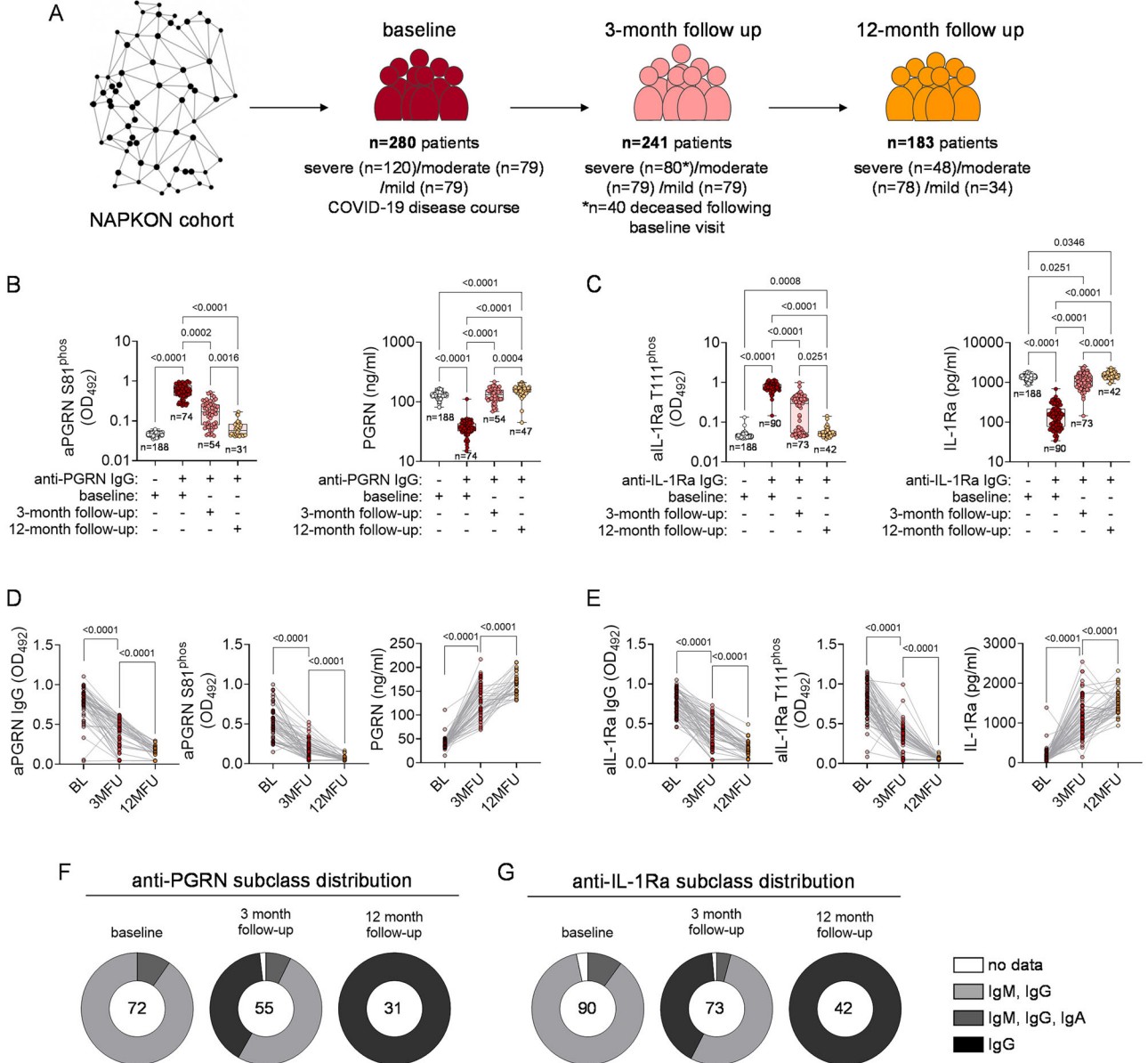

**Fig. 3 | Decline of hyperphosphorylated antigen precedes loss of seropositivity.** **A** Schematic overview of patients' samples collected via a national pandemic multi-center network (NAPKON, validation cohort II) and included in long term follow-up analysis of autoantibody status. The logo in Fig. 3A was reused with permission from NAPKON. **B,C** Presence of hyper-phosphorylated PGRN (**B**, left panel) and IL-1Ra (**C**, left panel) as well as endogenous PGRN (**B**, right panel) and IL-1Ra (**C**, right panel) levels were determined by in-house (phospho-antigens) or commercial ELISA (endogenous PGRN and IL-1Ra) in baseline and follow-up samples collected within NAPKON. Data are shown as box and whiskers plots (min-max), including all individual data points, and data were analyzed by Kruskal-Wallis followed by Dunn's

multiple comparison test. (**D, E**) Paired analysis of total anti-PGRN IgG, hyperphosphorylated PGRN and PGRN plasma levels (**D**, left to right; baseline (BL) to 3-month follow up (3MFU): $n = 62$; 3MFU to 12-month follow up (12MFU: $n = 36$) and total anti-IL-1Ra IgG, hyperphosphorylated IL-1Ra and PGRN plasma levels (**E**, left to right; BL to 3MFU: $n = 78$; 3MFU to 12MFU: $n = 50$) in baseline and follow-up samples collected within NAPKON. Data are shown as aligned dot plots with lines connecting paired observations and were analyzed by a two-tailed Wilcoxon signed-rank test. **F,G** Anti-PGRN (**F**) and anti-IL-1Ra (**G**) antibody subclass distribution in baseline and follow-up samples collected within NAPKON.

## COVID-19 survivors' monocytes hyperphosphorylate PGRN and IL-1Ra upon inflammatory stimulation

As our collective data imply a direct relationship between hyperphosphorylation of PGRN and IL-1Ra, and we demonstrate a correlation between hyperphosphorylated antigen levels and several inflammatory parameters (Fig. 5), we aimed to identify a more direct link between inflammatory load and excessive phosphorylation of IL-1Ra and PGRN. Initially, we observed that upon incubation of NB4 and U937 cell lines with selected proinflammatory stimuli, TNF and IFNγ, but not LPS, could induce hyperphosphorylation of IL-1Ra and PGRN (only IFNγ in NB4 cells) (Supplementary Fig. 15).

Subsequently, primary monocytes were isolated from healthy controls as well as convalescent survivors of critical COVID-19 12 to 14 months following severe disease episodes (Fig. 5A, B). COVID-19 patients were seropositive for both anti-IL-1Ra and anti-PGRN antibodies during severe disease but were seronegative at the time of monocyte isolation. Healthy control and (ex)patients' monocytes were stimulated in vitro with proinflammatory cytokines (IL-6, INFα2, INFγ) at different concentrations (Fig. 5A, B). In healthy donor monocytes, we observed that cell stimulation with the maximum tested concentration of IL-6 and IFNγ (10 ng/mL) resulted in hyperphosphorylation of IL-1Ra in cells from three (IL-6) or four (IFNγ) out of five tested donors (Fig. 5A). In these

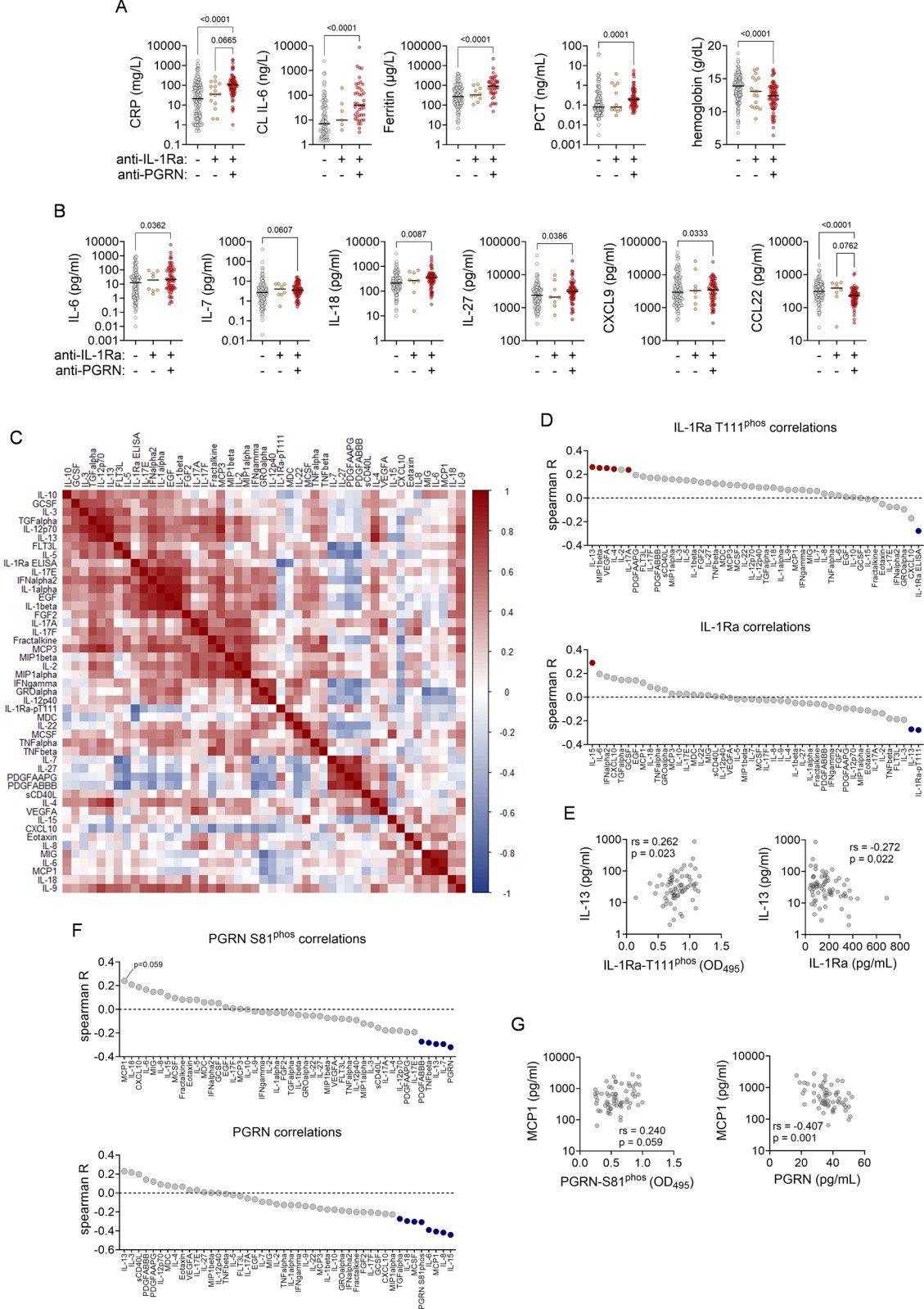

experiments, we observed no generation of PGRN isoforms (Fig. 5A). In contrast, in monocytes isolated from exCOVID-19 patients', we observed that already minor concentrations of all tested cytokines (IFNα2, IFNγ: 10 pg/mL; IL-6: 100 pg/mL) resulted in hyperphosphorylation of IL-1Ra (Fig. 5B). Hyperphosphorylation of PGRN only occurred upon monocyte stimulation with IFNα2 or IFNγ and required cytokine concentrations ≥ 1000 pg/mL(Fig. 5B).

## Discussion

In the context of SARS-CoV-2 infection several studies already reported autoantibody responses directed against several self-antigens[10,11,39,40]. In severe courses of MIS-C as well as mRNA vaccination-associated myocarditis, we previously reported on auto-antibodies targeting anti-inflammatory mediators, canonically antagonizing IL-1 (IL-1 receptor antagonist, IL-1Ra) or TNF-signaling

**Fig. 4 | Autoantibody seropositivity and presence of hyperphosphorylated antigen associates with (hyper)inflammatory mediator signatures and selected cytokine signaling. A,B** Clinical laboratory parameters (**A**) and markers from an experimental 45plex bead array assay (**B**) with significantly different levels in severe COVID-19 NAPKON baseline samples with indicated anti-PGRN and/or anti-IL-1Ra seropositivity (seronegative: $n = 135$; seropositive for only anti-IL-1Ra: $n = 8$; seropositive for both anti-IL-1Ra and anti-PGR: $n = 63$). Data are depicted as scatter dot plots with horizontal lines indicating respective median. Data were analyzed by Kruskal-Wallis followed by Dunn's multiple comparison test. **C** Multiple correlation analysis of inflammatory mediator levels determined by 45plex bead array assay and in-house ELISA (IL-1Ra, T111phos IL-1Ra) in seropositive severe COVID-19 NAPKON baseline samples. **D** Multiple association analysis of inflammatory mediator levels determined by 45plex bead array assay and in-house or commercial ELISA (IL-1Ra, T111phos IL-1Ra) with the presence of hyperphosphorylated IL-1Ra (**D**, upper panel;

T111phos IL-1Ra) and endogenous IL-1Ra levels (**D**, lower panel) in severe COVID-19 NAPKON baseline samples. **E** Spearman correlation and two-tailed significance test of correlation of IL-13 plasma levels with presence of hyperphosphorylated IL-1Ra (**E**, upper panel; T111phos IL-1Ra; $n = 71$) and endogenous IL-1Ra levels (**E**, lower panel; $n = 71$) in severe COVID-19 NAPKON baseline samples. **F** Multiple association analysis of inflammatory mediator levels determined by 45plex bead array assay and in-house or commercial ELISA (PGRN, S81phos PGRN) with the presence of hyperphosphorylated PGRN (**F**, left panel; S81phos PGRN) and endogenous PGRN levels (**F**, right panel) in severe COVID-19 NAPKON baseline samples. **G** Spearman correlation and two-tailed significance test of correlation of MCP-1/CCL2 plasma levels with presence of hyperphosphorylated PGRN (**G**, left panel; S81phos PGRN; $n = 63$) and endogenous PGRN levels (**G**, right panel; $n = 63$) in severe COVID-19 NAPKON baseline samples.

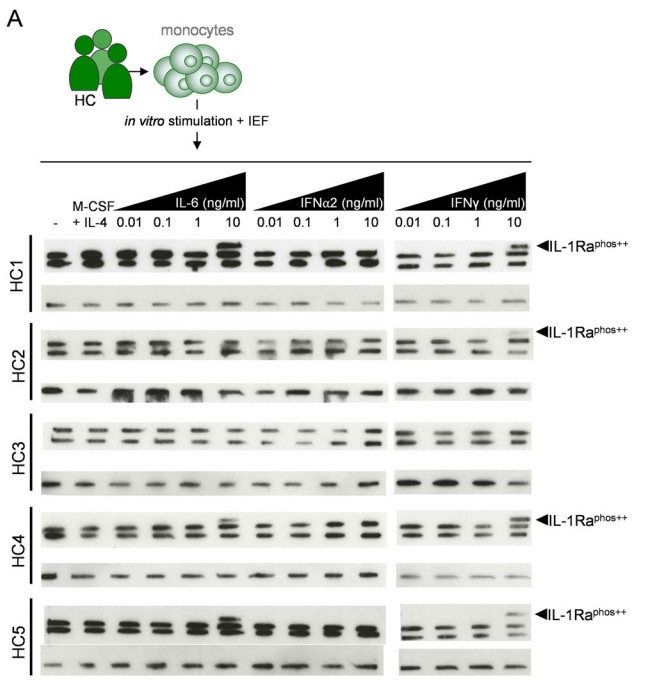

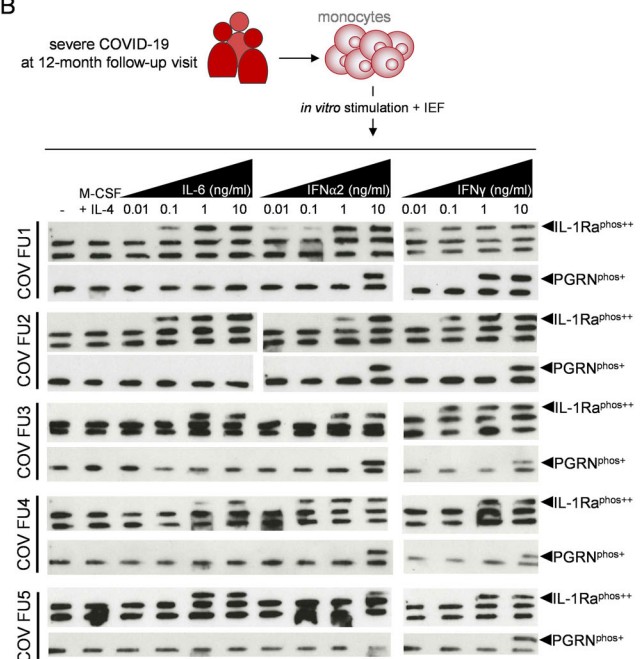

**Fig. 5 | COVID-19 survivors' monocytes hyperphosphorylate PGRN and IL-1Ra upon inflammatory stimulation. A,B** Monocytes were isolated from healthy control (**A**; HC, $n = 5$) or convalescent ex-COVID-19 patients (**B**; $n = 5$, previously diagnosed with severe disease and tested seropositive for both anti-PGRN and anti-

IL-1Ra antibodies). Cells were stimulated ex vivo with indicated recombinant cytokines for 48 h at 37 °C. Cell lysates were tested on IEF for the presence of phosphorylated PGRN (PGRNphos+) or hyperphosphorylated IL-1Ra (IL-1Raphos++).

(progranulin, PGRN). Yet, these studies have been limited by patient/sample sizes and lacked broadened functional analysis[19,20].

Here, we report on the occurrence of autoantibodies targeting both PGRN and IL-1Ra in a considerable proportion of adult patients with severe COVID-19. Importantly, such autoantibody responses were not or barely detectable in samples of moderate, mild or asymptomatic COVID-19 or in pre-pandemic septic patients with a pulmonary focus or in healthy controls. As also observed in previous studies, seropositivity for anti-IL-1Ra or anti-PGRN antibodies coincided with hyperphosphorylation of endogenous antigen, formation of antibody:self-antigen immune complexes and impairment of their anti-inflammatory bioactivity[18–20,29]. Upon transfection of seropositive COVID-19 patients' monocytes with respective constructs, alanine scanning enabled us to identify threonine 111 as the hyperphosphorylation site within IL-1Ra. We identified phospho-kinase (PK) A to drive canonical IL-1Ra phosphorylation in serine 97, whereas hyperphosphorylation in threonine 111 was mediated by PKC, and dephosphorylation by protein phosphatase 2A (PP2A). With respect to phosphorylation of serine 81 within PGRN[29], in previous work we

already identified PKCß1 and PP1 as the responsible kinase and phosphatase[29]. Using phage-display selected antibodies with specificity for hyperphosphorylated IL-1Ra or PGRN enabled us to track the prevalence of phospho-antigens and respective autoantibodies in a German national pandemic network (NAPKON) COVID-19 cohort. If seropositive, most analyzed patients were double-positive for both autoantibodies. Phospho-antigen levels and respective autoantibody titers were highest at the COVID-19 baseline, and both declined over the following 12 months, with a decrease in phospho-antigens preceding the decline in autoantibody concentrations. At baseline, autoantibody seropositivity was associated with laboratory markers for (hyper)inflammation and several inflammatory cytokine levels, which normalized in the course of follow-up. Multiplexed cytokine/chemokine profiling of patients' plasma allowed to associate the occurrence of hyperphosphorylated IL-1Ra and PGRN with specific inflammatory mediator profiles over time, but revealed no link with whole blood gene expression. Finally, we demonstrate that signaling by key inflammatory cytokines can induce IL-1Ra and PGRN hyperphosphorylation in isolated human monocytes. Importantly, in monocytes

isolated from previously seropositive survivors 12 to 14 months after severe COVID-19, this occurred at up to 1000-fold lower cytokine levels than in matched healthy controls.

Autoantibody responses targeting IL-1Ra have been previously reported by us and others with mixed results[18,20,21,41–46]. Utilizing an in-house ELISA platform based on HEK-cell expressed recombinant antigen combined with biochemical ex vivo detection of immune complexes and paralleling quantification of IL-1Ra serum/plasma levels using a commercial ELISA, we managed to identify such antibodies in MIS-C, SARS-CoV-2 mRNA vaccine-associated myocarditis, as well as Still's disease[18–20]. In contrast to the present data demonstrating a frequent co-occurrence of anti-IL-1Ra and anti-PGRN antibodies in severe COVID-19, in our previous studies in SARS-CoV-2 context, we detected anti-PGRN antibodies only in one out of 21 MIS-C patients, and in none of the investigated mRNA vaccination-associated endo-myocardial biopsy-confirmed myocarditis cases[19,20]. PGRN-specific autoantibodies were first identified in patients presenting with primary small, middle and large vessel vasculitis[34]. Of note, critical forms of COVID-19 have been shown to frequently associate with vasculitis-like characteristics[1], which may contribute to some of the disease-related complications.

Importantly, we now demonstrate that anti-IL-1Ra and anti-PGRN antibodies in severe COVID19 reveal no cross-reactivity with the other antigen. Beyond, we did not observe antibody binding to structural SARS-CoV-2 proteins, which may exclude molecular mimicry involving those proteins in driving the reported autoantibody responses.

Next to our own work, other studies successfully reporting on the occurrence of anti-IL-1Ra autoantibodies utilized cytokine microarray screening of plasmablast-derived mAbs[21], phage immunoprecipitation-sequencing (PhIP-seq) screening of autoantibody repertoires on peptide library combined with immunoprecipitation of radiolabelled full-length IL-1Ra by patient serum[46], or PhIP-seq combined with immunoprecipitation mass spectrometry (IP-MS)[44]. Importantly, the epitope specificity of anti-IL-1Ra antibodies identified in IgG4-related disease[21], pre-term delivery[44], MIS-C and Still's disease[18,19] appears to overlap, arguing for a rather narrow epitope specificity of such antibodies (Supplementary Fig. 16).

In contrast, studies using other approaches for auto-antibody detection[41,42,45], or used Luminex to quantify peripheral IL-1Ra plasma or serum levels[43] could not or only scarcely detect anti-IL-1Ra antibodies or observed no differences in serum/plasma antigen levels. From our own data we observed that the performance of our in-house ELISA to capture the described autoantibody responses is highly dependent on the source, quality and also storage conditions of the respective recombinant antigens. Moreover, in our hands commercial ELISAs as used in previous[18–20] as well as the present study clearly outperformed Luminex in spike recovery of recombinant IL-1Ra and PGRN as well as in detecting a depleting effect upon addition of either commercial or purified anti-IL-1Ra or anti-PGRN IgG. In the context of all immune assay formats applied by us and others, it is necessary to stress that endogenous immune complexes may sequester the free analyte (antigen or antibody) and can alter binding kinetics in such assay systems. Immune complexes may mask epitopes required for capture and/or detection reagent binding or may bridge assay antibodies and thus lead to under- or over-estimation of analyte concentration[47]. In this respect, we consider our approach of combining a biochemical assessment of immune-complexed and unbound antigen using native gradient gels and western blotting with immune assays (ELISA) a strength of our work.

In line with our previous findings[18–20,29], we observed seropositivity for anti-IL-1Ra and anti-PGRN antibodies to coincide with hyperphosphorylation of respective antigen, also in the context of severe COVID-19. However, in the present study, we also tracked the presence of hyperphosphorylated IL-1Ra and PGRN in patients' plasma from a multicenter national pandemic cohort over time. Utilizing

phage-display selected Fab-fragments with specificity for threonine 111 phosphorylated IL-1Ra and serine 81 phosphorylated PGRN, we observed the highest phospho-antigen levels and respective auto-antibody titers at the COVID-19 baseline. During follow-up, both declined, with a decrease in phosphoantigens preceding the decline in autoantibody titers. Importantly, competition experiments of selected Fabs against patients' IgG revealed that plasma antibodies against IL-1Ra or PGRN target both a phosphorylated epitope, but also unphosphorylated antigenic determinants.

On an anecdotal note, one patient enrolled in the discovery cohort was tested for anti-PGRN and anti-IL-1Ra antibodies within a 7-day interval before and after one cycle of plasmapheresis. This resulted in a rapid drop of autoantibody titers (Supplementary Fig. 17A–C), and paralleling recovery of IL-1Ra plasma levels (Supplementary Fig. 17D). In course of plasmapheresis hyperphosphorylated IL-1Ra disappeared, while the hyperphosphorylated isoform of PGRN remained detectable (Supplementary Fig. 17E). Even though we have not yet formerly demonstrated, that hypherphosphorylated antigen can break peripheral tolerance, these data may further strengthen such causal link. In other contexts reported previously[38,48–50], we already managed to identify phospho-antigen specific T cell responses[51], while phosphorylated T cell epitopes have been assigned high MHCI/II affinity and to induce strong CD4 and CD8 T cell responses[52,53]. In case of anti-PGRN and anti-IL-1Ra antibodies as described in the present study, the transience of the antibody response may also argue for the involvement of an extra-follicular B cell activation, resulting in the formation of rather short-lived antibody-producing plasma cells and no establishment of B-cellular memory.

Extrafollicular B cell responses can be driven and supported by inflammatory stress[54,55]. Our data indicate, that seropositivity for anti-IL-1Ra and anti-PGRN antibodies coincides with elevation in clinical laboratory markers for (hyper)inflammation and several inflammatory cytokines. Further, and in an important difference to previous observations in MIS-C or mRNA vaccination-associated myocarditis, severe COVID-19 patients frequently presented with both anti-IL-1Ra and anti-PGRN antibodies and double seropositivity associated with the highest inflammatory marker levels. In this respect, the decline in CCL22 in double seropositive individuals is interesting. While CCL22 has several functions in macrophage polarization but also regulatory T cell attraction, anti-CCL22 antibodies have been described as part of a COVID-19 specific anti-chemokine autoantibody signature[40].

Beyond, we observed in multiple correlation analysis that IL-13 plasma levels were significantly associated with hyperphosphorylation of IL-1Ra and paralleled a decrease in IL-1Ra plasma levels. Of note, blood IL-13 levels have been previously linked with COVID-19 severity and need for mechanical ventilation, while therapeutic IL-13 blockade prevented COVID-19 in SARS-CoV-2 susceptible mice[56]. In the course of follow-up, the association with IL-13 was lost, and instead we observed significant associations of IL-8 as well as FLT3L with both IL-1Ra hyperphosphorylation as well as peripheral IL-1Ra depletion. IL-8 has been reported by us and others in context with COVID-19 severity[4], while FLT3L has been demonstrated to have a crucial role in B cell differentiation and generation of antibody responses[57].

With PGRN hyperphosphorylation and paralleling antigen depletion in peripheral blood at baseline and during follow-up we only observed partially significant association with MCP-1/CCL2 and IL-27, respectively. MCP-1/CCL2 has been described to link with COVID-19 severity, deteriorated progression and fatal outcome[58], while IL-27 – a member of the IL-12 cytokine family - has been demonstrated to associate with anti-spike protein antibody titers in hospitalized COVID-19 patients[59].

When analyzing whole-blood transcriptomics among COVID-19 patients enrolled in NAPKON according to anti-IL-1Ra/anti-PGRN status, the observed prominent association of seropositivity for any of the

autoantibodies with platelet activation and blood coagulation pathways is likely just a reflection of the autoantibodies' strong association with severe disease. High disease burden in COVID-19 was associated already previously with a prominent platelet activation and disturbed blood coagulation gene and protein expression signature[60–62]. When we compared severely ill COVID-19 patients seropositive for autoantibodies with equally sick COVID-19 patients but seronegative for anti-PGRN/anti-IL-1Ra, we observed no significant differences in gene expression among whole blood cells. Yet, this analysis was likely affected by a reduced number of samples and high variability. To adequately address the question of a causal link of blood cell gene expression, hyperphosphorylation and anti-IL-1Ra/anti-PGRN autoantibodies, an approach involving RNAseq of monocytes from previously severely ill and seropositive (or negative) for anti-PGRN/anti-IL-1Ra antibodies and re-challenged in vitro with inflammatory cytokines to induce hyperphosphorylation may provide more conclusive data. Furthermore, to better understand the role of underlying genetics, we are currently in the process of assembling data sets from the NAPKON registry alongside with serological autoantibody status, but this is yet beyond the scope of the present work.

Collectively, in severe COVID-19, our data points to an association of anti-IL-1Ra and anti-PGRN antibodies with the load of inflammation. On the level of tissue-infiltrating T cells and macrophages, as well as markers of cardiac stress and damage, we previously reported similar findings in association with seropositivity for anti-IL-1Ra antibodies in the context of myocarditis following mRNA vaccination against SARS-CoV-2[20]. However, on the basis of these data, it remains difficult to determine whether the observed inflammatory manifestations are cause or rather consequence of the IL-1Ra and PGRN-targeting antibodies. Importantly, in the present study we now demonstrate that in vitro stimulation of monocytes isolated from healthy controls with selected inflammatory cytokines resulted in hyperphosphorylation of IL-1Ra in some tested donors. In contrast, paralleling cytokine stimulations of cells isolated from critical COVID-19 survivors (12-14 months post-severe disease), who had tested seropositive for anti-IL-1Ra and anti-PGRN autoantibodies in the course of their severe disease episodes, showed that compared to healthy controls already minor cytokine concentrations (up to 1000-fold less) were required to induce particularly IL-1Ra hyperphosphorylation. Hyperphosphorylation of PGRN was only observed upon higher concentrations of type I/II interferon stimulation. Together, this suggests that excessive inflammation can drive particularly IL-1Ra hyperphosphorylation, which may be in line with the observation that we frequently observed anti-IL-1Ra and anti-PGRN double-positive or anti-IL-1Ra single-positive patients with COVID-19, but no patients presenting with anti-PGRN antibodies exclusively.

Altogether and based on the present data we currently hypothesize that inflammation itself can drive transient hyperphosphorylation of IL-1Ra and PGRN, which may then result in a break of peripheral tolerance at the B and T cell level, depending on a genetic predisposition of individuals, which remains yet to be defined. Furthermore, our observations on the impact of inflammatory stress (i.e., mediated by signaling of prominent inflammatory cytokines) and induction of hyperphosphorylation suggest that this, and related autoantibody responses, may not be limited to a SARS-CoV-2 infection and linked hyperinflammatory context. This is supported by data from us and others reporting on anti-IL-1Ra and/or anti-PGRN antibodies in inflammatory conditions not associated with SARS-CoV-2[18,21,34,44,63].

Altogether, our findings need to be interpreted in the light of several limitations. First, we lack genomic data regarding study patients. This leaves important questions regarding the potential association of genomic variants with the susceptibility to hyperphosphorylation of IL-1Ra and PGRN upon inflammatory signaling and autoantibody generation yet unanswered. Second, we also lack immunophenotyping data on cellular level. While for some patients enrolled in this study respective biomaterial was available (as used for i.e., in vitro induction of hyperphosphorylation), for the large majority of study patients, we only had access to serum or plasma samples. Third, while our functional experiments on induction of hyperinflammation involve hallmark inflammatory cytokines, such as IL-6, IFNα and IFNγ, we did not specifically test other potential candidates suggested from our multiple association analysis as i.e. IL-13, MCP-1/CCL2 or IL-27.

Nevertheless, we believe that our present analysis and data advance our understanding regarding the prevalence of anti-IL-1Ra and anti-PGRN antibodies in context of severe SARS-CoV-2 infection. For the first time, we provide robust evidence for a link between autoantibody seropositivity and the level of inflammation and demonstrate a role for inflammatory cytokine signaling as a trigger of IL-1Ra and PGRN hyperphosphorylation.

## Method

### Study population

The study was approved by the local Ethical Review Board (Ethics Committee of Saarland University, Homburg, Germany; Bu 62/20). Plasma samples of discovery, validation and control cohorts were locally obtained from the CORSAAR registry study (Saarland, Germany). From a few patients enrolled in CORSAAR, we also prepared PBMCs. All patients or their legal representatives provided written informed consent. Analysis for proinflammatory autoantibodies and in-depth characterization in samples of CORSAAR and other trials was also approved by the local Ethical Review Board (Ethics Committee of Saarland University, Homburg, Germany; Bu42/21). These other trials/registries comprised plasma samples obtained from convalescent individuals enrolled in the CAPSID trial (EUDRA-CT 2020-001310-38), samples of the CoKiBa study (research project of COVID-19 in children and adolescents by a network of local practicing pediatricians and the University Hospital Regensburg obtained 1-3 months after the peak of the first pandemic wave in spring 2020 in Bavaria, Germany), samples of patients with severe, moderate and mild COVID-19 enrolled in NAPKON (National Pandemic Cohort Network, Germany), and pre-pandemic infectious control samples of the MAXSEP (NCT00534287) and the SISPCT trial (NCT00832039) of the German Sepsis Network. Please see Supplementary Fig. 1 for numbers (n) and the use of these samples in the context of the present study. All participants or their legal representatives provided written informed consent to the collection of blood samples and clinical data as part of these studies or registries. All plasma and PBMC samples were stored at −20 °C or −80 °C, respectively, until use.

### ELISA for plasma antibodies targeting different self-antigens

The ELISA for autoantibodies was performed as previously described[34]. In short, the antigens were obtained using the coding sequences of the *GRN* gene encoding PGRN, IL-10 and isoform 1 precursor and isoform 2 of *IL1RN*, and the fragments aa1-63, aa107-177 and aa47-123 of isoform 1 of IL-1Ra, IL-18BP, IL-22BP, IL-36-Ra, CD40, serpinB1, IFNα2, IFNγ and IFNω, which were recombinantly expressed with a C-terminal FLAG-tag in HEK293 cells under the control of a cytomegalovirus promoter (pSFI). Total cell extracts were prepared and bound to Nunc MaxiSorp plates (ThermoScientific, Frankfurt, Germany, Prod. No 442404) pre-coated with murine anti-FLAG mAb at a dilution of 1:2500 (v/v; Sigma-Aldrich, Munich, Germany, Prod. No. F3165) at 4 °C overnight. After blocking with 1.5% (w/v; 1 h RT) gelatin in Tris-buffered saline (TBS) and washing steps with TBS with Triton X-100, the individual plasma samples were diluted 1:100 and added for 1 h at RT. Following repeated washing steps and depending on the scientific question, different detection antibodies were added for 1 h at RT (biotinylated goat anti-human heavy and light chain immunoglobulin G (IgG; 1:2500; Dianova,

Hamburg, Germany, Prod. No. DNA-SEC183078); subclass-specific sheep anti-human IgG1, IgG2, IgG3 and IgG4 (all 1:5000; Binding Site Group, Birmingham, UK, Prod. Nos. AU006, AU007, AU008, AU009); goat anti-human IgM (1:2500; Dianova, Prod. No. DNA-SEC183006); goat anti-human IgA (1:2500; Dianova, Prod. No. DNA-SEC183007). Following this step, corresponding biotinylated or peroxidase-labeled secondary antibodies were used for immunoassays carried out to detect IgG subclasses (anti-sheep IgG-POX, Sigma Aldrich, Prod. No. A9452; 1:5000, 1 h RT) and IgA/M (biotinylated anti-goat IgG, Dianova, Prod. No. DNA-SEC182876; 1:2500, 1 h RT). Assays were developed by addition of peroxidase-conjugated streptavidin (StreptPOX, 1:50.000, Roche Applied Science, Indianapolis, IN, USA, Prod. No. 11089153001; 10 min, RT) and OPD (Sigma Aldrich, Prod. No. P5412). As a cut-off for positivity, the average of the optical density (OD) of the negative samples plus three standard deviations was applied.

### Analysis of IL-1Ra- or PGRN-immunocomplexes

For detection of immune complexed PGRN and IL-1Ra in patients' plasma (1:100, 15 μl) of samples were run on native gradient PAGE (4-20%, without reducing sample preparations and without SDS) were performed, followed by western blotting (tank transfer, 1 h 100 V) onto PVDF membranes. Membranes were blocked (o/n, 4 °C in TBST/milk buffer (10% [v/v] milk in 10 mmol/L TrisHCl, pH7·5, 150 mmol/L NaCl, and 0·1% [v/v] Tween 20), washed and incubated (1 h, RT) incubated with either murine anti-hPGRN antibody (abcam, Prod. No. ab169325, 1:2000) followed by anti-mouse/POX (Biorad, Prod. No. 170-6516, 1:3000), rabbit anti-hIL-1Ra antibody (antibodies-online, Prod. No. ABIN2856394, 1:2000) followed by anti-rabbit/POX (Biorad, Prod. No. 170-6515, 1:3000), or biotinylated goat anti-human IgG (F(ab')2 antibody (1:2500; Dianova, Prod. No. 109-066-097). All incubation steps were carried out for 1 h at RT and following repeated washing steps in TBST. Finally, blots were incubated with Strep/Pox (Roche, Prod. No. 11089153001; 1:15000), washed and developed using ECL reagents.

To isolate PGRN and IL-1Ra immunocomplexes, mouse anti-human PGRN antibodies (abcam, Prod. No. ab169325) or rabbit anti-human IL-1Ra antibodies (antibodies-online, Prod. No. ABIN2856394) or mouse anti-human PGRN antibodies (abcam, Prod. No. ab169325) at 5 μg/mL were incubated and coupled with Affi-gel Hz Hydrazide (Biorad, Prod. No. 156-0016; o/n, 4 °C). Plasma (1:100 dilution) of one patient with critical COVID-19 (Val 51) and seropositive for anti-PGRN and anti-IL-1Ra antibodies and one patient (Val 52) with critical COVID-19 but seronegative for anti-PGRN and anti-IL-1Ra antibodies, were incubated at 4 °C overnight with either Affi-matrix with rabbit anti-human IL-1Ra antibodies or with mouse anti-human PGRN at 5 μg/mL. Elution was performed using 0.25 M glycine (pH 3.0). Eluted proteins were separated by SDS-PAGE. Western Blots (semidry, 1 h, PVDF membrane) were performed with anti-IL-1Ra and anti-PGRN antibodies and developed as described above.

### Gel filtration

In order to characterize anti-IL-1Ra and anti-PGRN immune complexes in patients' plasma, we determined the molecular weight of the different components by FPLC and gel filtration. For calibration of the Superdex 200 PrepGrade HL16/60 column alpha-Lactalbumin (14.4 kDa), bovine Albumin (66 kDa), equine Transferrin (76 kDa), alpha-2-Makroglobulin (170 kDa), bovine Catalase (232 kDa), Ferritin (440 kDa), Thryoglobulin (669 kDa), and Dextran 1000 (1000 kDa) were used (LMW-Kit: GE, Prod. No. 28-4038-41 and HMW-Kit Prod. No. 28-4038-42). Individual protein peak fractions were collected and were subjected to native gradient PAGE and western blotting as described above.

### Artificial immune complexes

Artificial IL-1Ra or PGRN immune complexes were generated by incubating recombinant human IL-1Ra or PGRN (both 1 μg; antibodies online, Prod. Nos. ABIN1111703 or ABIN5564456) with monoclonal or polyclonal anti-human IL-1Ra or anti-human PGRN antibodies (all at 1 μg; monoclonals: antibodies online, Prod. Nos. ABIN7425773 and ABIN516191; polyclonals: antibodies online, Prod. Nos. ABIN2856394 and ABIN11169131). Resulting artificial immune complexes were subjected to native gradient PAGE and western blotting as described above.

### Isoelectric focusing (IEF) and immunoblotting of PGRN and IL-1Ra

For IEF of plasma PGRN and IL-1Ra, plasma was pre-diluted 1:100 in 1xPBS, and the mixed with IEF sample buffer at a ratio of 1:2. Samples were analyzed by IEF on a gel with a fixed pH gradient (pH 3–10) according to the manufacturer's instructions (Novex pH 3–10, Invitrogen, Germany, Karlsruhe) followed by immunoblotting. Western blotting and IEF of intracellular proteins was performed as already described[38]. Immunoblotting of IEFs was performed by semi-dry blotting (1 h) onto PVDF membranes. Blots were incubated with anti-IL-1Ra and anti-PGRN antibodies and developed as described above.

### Phosphatase treatment and IEF of plasma IL-1Ra

Plasma of indicated patients was diluted 1:10 in phosphatase-specific buffer (Fermentas/VWR, Darmstadt, Germany, Prod. No. EF0651). Samples were incubated with FastAP thermo-sensitive alkaline phosphatase (1U/μL; Fermentas/VWR, Darmstadt, Germany, Prod. No. EF0651)1 U/μL per 500 μL lysate) at 37 °C overnight. Phosphatase was inactivated by heating samples to 80 °C (10 min). Equal volumes of sample and loading buffer were mixed and analyzed by IEF and immunoblotting as described above.

### Site-directed mutagenesis

Using QuickChange II Site-Directed Mutagenesis Kit (Stratagene) and full-length *IL1RN* DNA fragments with C-terminal FLAG tag, all candidate phosphorylation sites (serine, threonine, tyrosine) outside of the promoter region were considered for mutagenesis. Serine (S) and threonine (T) were replaced by alanine (A) and tyrosine (Y) was replaced by phenylalanine (F). Specifically, within *IL1RN* (cloned via *EcoR*V in pSfi-FLAG; G. Bornkamm, German Research Center for Environment and Health, Munich) the following constructs were generated: S28A, S32A, S33A, T47A, Y49F, Y59F, S93A, S97A, T101A, T111A, S114A, S128A, S130A, T133A, T134A, S135A, S138A, T148A, S157A, T159A, T169A and Y172F. FLAG-tagged mutants were cloned via *Sfi*I into pRTS1 (G. Bornkamm, German Research Center for Environment and Health, Munich). Resulting vectors were introduced into primary human monocytes isolated by CD14 MACS (Pan Monocyte isolation kit, Miltenyi, Bergisch Gladbach, Germany, Prod. No. 130-096-537) by electroporation (Biorad GenPulser, 160 V in 0.2 μm cuvette, 100 μl cell suspension at 1x10⁷ cells/ml in plain RPMI1640). Monocytes were isolated from a patient who tested positive for hyperphosphorylated IL-1Ra and anti-IL-1Ra antibodies and a control without hyperphosphorylated IL-1Ra and no IL-1Ra antibodies. Following electroporation cells were cultured in RPMI (PANBioTech, Aidenbach, Germany, Prod. No. P04-16500) and 20% FCS with M-CSF at 10 ng/ml and IL-4 at 1 ng/ml (added every 48 hours; Peprotech, Hamburg, Germany) for 5 d. Cell lysates were analyzed by IEF and western blotting as described above.

### Phage display selection of T111^phos IL-1Ra-specific Fab fragments

The generation of Fab-fragments with specificity for hyperphosphorylated (T111^phos) IL-1Ra or IL-1Ra with physiological phosphorylation (S97^phos) was performed according to phage display protocols as described earlier in the context of PGRN[38]. For phage display selections N-terminally biotinylated IL-1Ra fragments spanning amino acids 86 to 107 or 101 to 124 were generated as synthetic peptides (GeneCust, Luxembourg) either phosphorylated or non-phosphorylated at Serine 97 or Threonine 111, respectively (IL-1Ra 86-107: bio-SGGGGS

HGGKMCLSCVK S GDETRLQLEA; IL-1Ra 86-107 with S97[phos]: bio-SGGGGS HGGKMCLSCVK S[phos] GDETRLQLEA; IL-1RN 101-124: bio-SGGGGS TRLQLEAVNI T DLSENRKQDKRFA; IL-1RN 101-124 with T111[phos]: bio-SGGGGS TRLQLEAVNI T[phos] DLSENRKQDKRFA). Purity (<95%) of synthetic peptides was verified by HPLC (GeneCust, Luxembourg).

As a phagemid library, a non-immune, semi-synthetic human Fab repertoire containing $3.7 \times 10^{10}$ individual antibody fragments was utilized. Phage particles were pre-incubated for 1 h at room temperature (RT) in 2% non-fat dry milk/PBS. Subsequently, phage particles were incubated for 45 min with decreasing concentrations of biotinylated phosphorylated or biotinylated non-phosphorylated IL-1Ra peptides (300, 200 and 100 nM). Streptavidin beads were added and incubated with phages and peptides with constant rotation (15 min, RT). Incubation was followed by six wash cycles with 2% non-fat dry milk plus 0.1% Tween, followed by six cycles with 0.1% Tween-PBS and by two additional washes with 1xPBS. Finally, bound phages were eluted with 100 mM Triethylamin and samples were neutralized by addition of Tris-HCl (pH 7.2). Eluted phage clones were used to infect a bacterial host strain (E. coli, TG1; 30 min at 37 °C), and bacteria were grown overnight at 30 °C on plates. Following this amplification step, phage particles were subjected to the next round of selection. In the second and third rounds of selection, phages were pre-incubated with streptavidin-coated paramagnetic beads (Invitrogen and Dynal, Oslo, Norway) to remove streptavidin binders. Moreover, in order to increase the probability to obtain phage clones discriminating between the IL-1Ra isoforms, phages coming from selections on S97 phosphorylated or T111 hyperphosphorylated IL-1Ra were pre-absorbed with the respective non-phosphorylated IL-Ra peptides (IL-1Ra 86-107 and IL-1Ra 101-124). Vice versa, phages coming from selections on non-phosphorylated peptides were pre-absorbed with S97[phos] and T111[phos] containing IL-1Ra peptides (IL-1Ra 86-107 with S97[phos] and IL-1Ra 101-124 with T111[phos]). To yield Fab fragments with reliable specificity for T111[phos] IL-1Ra three rounds of selection were necessary. For Fab expression, E. coli (TG1) were transformed with the phagemid vector pCES1 containing the Fab sequences for expression as His6-tagged proteins as described previously[64]. After lysis with PBS, Fabs were purified by IMAC chromatography (Qiagen), concentrated and stored at −20 °C until use. Specificity of Fabs for the non-phosphorylated or the phosphorylated IL-1Ra isoform was verified by IEF.

### ELISA for PGRN isoforms with and without phosphorylation in Ser81

To detect antibodies with specificity for S81[phos] PGRN, Nunc MaxiSorp plates were precoated with murine anti-HIS mAb at a dilution of 1:2500 (v/v; Qiagen, Hilden, Germany, Prod. No. 34660) at 4 °C overnight. After blocking with 1.5% (w/v) gelatin in Tris-buffered saline (TBS) and washing steps with TBS with Triton X-100, individual plasma samples were applied at a dilution of 1:2. For the detection of the hyperphosphorylated (S81) or the non-phosphorylated S81 PGRN isoform, S81[phos]-PGRN or non-phosphorylated S81 PGRN-specific Fabs, which had previously been selected by phage display screening[29]), were added at a concentration of 5 μg/ml (1 h, RT). Following repeated washing, biotinylated goat anti-human IgG (F(ab')2 antibody (Dianova, Prod. No. 109-066-097; 1:2500, 1 h RT) was added. After repeated washing and addition of peroxidase-labeled streptavidin (StreptPOX, 1:50.000; Roche, Prod. No. 11089153001) the assay was developed using OPD.

### ELISA for IL-1Ra isoforms with and without phosphorylation in Threonine 111

For ELISA for the T111[phos]-IL-1Ra isoform Nunc MaxiSorp plates were coated overnight (4 °C) with rabbit anti-human IL-1Ra (1:2000; anti-bodies-online, Prod. No. ABIN2856394), followed by blocking with 1.5%

(w/v) gelatin in TBS (1 h, RT) and washing steps using TBS-Tx [TBS, 0.1% (v/v) Tx100]. Individual plasma samples were applied (1:3, 1 h, RT). For the detection of the hyperphosphorylated (T111) or the non-phosphorylated T111 IL-1Ra isoform, T111[phos]-IL-1Ra or non-phosphorylated T111 IL-1Ra-specific Fabs selected by phage display screening as described above, were added at a concentration of 5 μg/ml (1 h, RT). Following repeated washing, biotinylated goat anti-human IgG (F(ab')2 antibody (1:2500, 1 h, RT; Dianova, Prod. No. 109-066-097) was added. After repeated washing and addition of peroxidase-labeled streptavidin (StreptPOX, 1:50.000; Roche, Prod. No. 11089153001) the assay was developed using OPD.

### Cell line and primary human monocyte stimulation and IEF

Lysates of A431 (DSMZ, Braunschweig, Germany, Prod. No. ACC91), FaDu (DSMZ, Prod. No. ACC784), Mewo (ATCC, Prod. No. HTB-65), Mel-Juso (DSMZ, Prod. No. ACC74), NB4 (DSMZ, Prod. No. ACC207) and U937 (DSMZ, Prod. No. ACC5) cell line were analyzed by western blot and IEF for expression of IL-1Ra and isoforms thereof. Subsequently, Mel-Juso, NB4 and U937 lines ($2 \times 10^5$ cells) were cultured in RPMI1640 + 10%FCS+glutamin (Sigma Aldrich, Prod. No. G7513 and stimulated by recombinant human TNF, IFNγ (Peprotech, Hamburg, Germany, Prod. Nos. 300-01 A and 300-02) or LPS (E. coli, O111:B4; Merck, Darmstadt, Germany) each at 10 ng/ml for 5 d. Cell lysates were prepared by the addition of 50 μl 8 M urea. 10 μl were loaded on gel for IEF analysis. Primary human monocytes from follow-up samples 12 to 14 months after critical COVID-19 of 5 survivors, who tested positive for anti-IL-1Ra- and anti PGRN-antibodies during critical disease but were seronegative at follow-up, and of age- and sex-matched healthy controls were isolated (Miltenyi, Monocyte isolation kit). Cells were cultured ($1 \times 10^5$ cells in RPMI1640 + 20%FCS+glutamin) with M-CSF (10 ng/ml) and IL-4 (1 ng/ml) and with or without recombinant human cytokines (IL-6, IFNγ or INFα2; all from Peprotech, Prod. Nos. 200-06, 300-0 and 300-02 K) at concentrations from 0.1 to 10 ng/ml for 5 d. Cell lysates were prepared by the addition of 50 μl 8 M urea. 10 μl were loaded on gel for IEF analysis.

### ELISA for PGRN and IL-1Ra plasma levels

PGRN and IL-1Ra plasma levels were determined using commercially available ELISA kits (PGRN: AdipoGen, Incheon, South Korea, Prod. No. AG-45B-0027; IL-1Ra: Invitrogen/ThermoFisher, Prod. No. BMS2080) according to the manufacturer's instructions.

### TNF-induced cytotoxic effect (MTT assay)

To assess the functional effects of PGRN antibodies in vitro, a non-radioactive viability assay (EZ4U Cell Proliferation Assay; Biomedica, Vienna, Austria, Prod. No. BI-5000) was performed. For this TNF-induced cytotoxicity indicator assay, we used the highly TNF-sensitive mouse fibrosarcoma WEHI-S cell line (DSMZ, Prod. No. ACC25) as target cells. In short, $4 \times 10^4$ WEHI-S cells were seeded into 200 μl of cell culture at 37 °C and 5% $CO_2$. To detect possible differences in TNF-inhibiting activity in plasma between patients with or without PGRN antibodies, plasma of patients with COVID-19 with or without PGRN antibodies was added in dilutions from 1:8 to 1:512 to cultured WEHI-S cells, followed by administration of TNF (100 pg/ml). In some experiments, Etanercept or Adalimumab at concentrations of 0.25 to 2.0 μg/ml were added to patients' plasma. WEHI-S cells without TNF or plasma, or with TNF (100 pg/ml) added alone, were used as negative and positive controls, respectively. After 48 h of incubation at 37 °C, chromophore substrate was added to each well. This chromophore substrate is converted only by vital cells. The adsorption of the product was measured at an OD of 450 nm. Due to the very short half-life of TNF compared to PGRN, and to exclude any possible unknown interferences from other plasma components, the MTT assay was repeated with TNF (100 pg/ml) and recombinant PGRN (10 ng/ml) with either recombinant PGRN antibodies, PGRN antibodies purified from

plasma of patient Val16 at concentrations from 0 μg/ml to 10 μg/ml or as a control SLP2-paraprotein purified from a patient with multiple myeloma at 10 μg/ml.

## IL-1β signaling reporter assay

For the IL-1β reporter assay, we used HEK-Blue™ IL-1β reporter cells (Invivogen, Prod. No. hkb-il1bv2), which respond to IL-1β and IL-1α signaling by induction of NF-κB/AP-1, leading to expression of a secreted embryonic alkaline phosphatase (SEAP) reporter. Anakinra (Kineret, Swedish Orphan Biovitrum) 40 ng/mL, recombinant IL-1Ra at 40 ng/mL (Biozol, Prod. No. PPT-AF-2000-01RA), recombinant IL-1Ra at 40 ng/mL (Biozol, Prod. No. PPT-AF-2000-01RA) and anti-IL-1Ra antibody at 5 μg/mL (antibodies-online, Prod. No. ABIN2856394), recombinant IL-1Ra at 40 ng/mL (Biozol, Prod. No. PPT-AF-2000-01RA ml and recombinant SLP-antibody at 5 μg/mL (abcam, Prod. No. ab191883), plasma diluted 1:20 of a patient with COVID-19 and high-titer IL-1Ra-antibodies (Val 16) with and without recombinant IL-1Ra at 40 ng/mL, and plasma diluted 1:20 of a patient with COVID-19 without high-titer IL-1Ra antibodies (Val 44) with or without recombinant IL-1Ra at 40 ng/mL were preincubated for 2 h at room temperature. Subsequently, these samples were added together with either recombinant human IL-1β (Biozol, Prod. No. PPT-200-01B) or TNF (Biozol, Prod. No. PPT-300-01A) both at 2 ng/mL in 100 μl DMEM to $2 \times 10^4$ HEK-Blue™ IL-1β reporter cells per well and incubated overnight at 37 °C. Thereafter, 180 μL of each supernatant was transferred, 20 μl QUANTI-BlueTM (Invivogen, Prod. No. rep-qbs) was added, and SEAP activity was measured at OD of 655 nm. Experiments were performed in triplicate.

## ELISA for cross-reactivity of purified anti-PGRN and anti-IL-1Ra antibodies against structural proteins of SARS-CoV-2

PGRN and IL-1Ra antibodies were enriched from plasma obtained from the patients listed below. For this purpose, 500 μl of lysates of HEK293 cells transfected with recombinant C-terminally FLAG-tagged PGRN or IL-1Ra were incubated with 20 ml anti-FLAG matrix (15 min, RT). Anti-PGRN- and IL-1Ra antibody positive patient's plasma (500 μl) were diluted 1:10 (v/v) in PBS and incubated with the anti-FLAG matrix/FLAG-tagged PGRN or IL-1Ra complex and subsequently eluted using 100 ml of 0.1 M glycine (pH 3.0). Patient plasma, flow-through and eluted enriched anti-PGRN or anti-IL-1Ra antibodies and plasma of controls listed below were screened by ELISA for reactivity against recombinant HIS-tagged SARS-CoV-2 S1- and S2-proteins, N-protein and M-protein (AGRO, Frankfurt, Germany, Prod. Nos. SIN-C52H3 and NUN-C5227; ProSci LLC, Poway, CA, USA, Prod. No. 39107; Chimerigen Lab, Liestal, Switzerland, Prod. No. CHIB233501) expressed in HEK293 cell, or tested for binding to FLAG-tagged PGRN, precursor of IL-1Ra isoform 1 and IL-1Ra isoform 2. PGRN antibodies were purified from plasma patient No. 10 and No. 20 of the cohort with moderate and severe COVID-19 infection, respectively. Plasma from two healthy control, from a patient with rheumatologic disease with PGRN antibodies and without COVID-19, from a patient with rheumatologic disease without PGRN antibodies and without COVID-19 served as controls.

## Competition ELISAs

To validate the specificity of IL-1Ra- and PGRN-antibodies in patients' plasma, antibodies were purified from the plasma of seropositive individuals using an Affi-matrix (see above). Nunc MaxiSorp plates were precoated with rabbit anti-FLAG at a dilution of 1:2500 (v/v; Sigma Aldrich, Prod. No. F7425) at 4 °C overnight. Following blocking with 1.5% (w/v; 1 h RT) gelatin in Tris-buffered saline (TBS) for 1 h at RT and washing steps with TBS with Triton X-100, either purified autoantibody or commercial murine anti-human PGRN (antibodies online, Prod. No. ABIN516191) or anti-IL-1Ra antibodies (antibodies online, Prod. No. ABIN7425773) were added at concentrations and combinations as indicated in Supplementary Fig. 5C−F. Following repeated washing steps either biotinylated goat anti-murine immunoglobulin G (IgG), 1:2500,

(Biozol, Prod. No. DNA-SEQ-183154) or biotinylated goat anti-human IgG (F(ab')2 antibody (1:2500; Dianova, Hamburg, Germany, Prod. No. 109-066-097) the ELISA was added, and assays were developed by addition of peroxidase-conjugated streptavidin (StreptPOX, 1:50.000 (Roche, Prod. No. 11089153001; 10 min, RT) and OPD.

For competition assays of purified anti-IL-1Ra antibodies against phage-display selected Fabs with specificity for hyperphosphorylated (T111phos) or non-hyperphosphorylated IL-1Ra, wtIL-1Ra or T111phosIL-1Ra were purified from patients' plasma and immobilized indirectly via rabbit-anti-human IL-1Ra (antibodies online, ABIN2856394) on Nunc MaxiSorp plates in coating buffer at 1:2000 dilution. Following blocking with 1.5% (w/v; 1 h RT) gelatin in Tris-buffered saline (TBS) for 1 h at RT and washing steps with TBS with Triton X-100, plates were simultaneously incubated with Fabs (0.5 μg/ml) and purified IgG (0-10 μg/ml). Followed by repeated washing steps, plates were developed upon incubation with mouse-anti- HIS antibody (1:2000; Qiagen, 34660) followed by biotinylated anti-mouse IgG antibody (1:2500; Dianova, DNA-SEC183803) and StreptPOX (1:50.000; Roche, Prod. No. 11089153001).

## Phosphorylation/Dephosphorylation of Threonine 111 in IL-1Ra

Panels of kinase or phosphatase inhibitors at indicated concentrations were administered to cultures ($2 \times 10^5$ cells/ml) of primary monocytes derived from patients with hyperphosphorylated IL-1Ra or from healthy controls. After 3 days cells were harvested, and lysates were analyzed by IEF and immunoblotting as described above.

## Coimmunoprecipitation of identified kinases and phosphatases with T111phos IL-1Ra

Monocyte cell lysates of one healthy control and of one patient tested seropositive for anti-IL-1Ra antibodies and hyperphosphorylated IL-1Ra were incubated at 4 °C overnight with a goat anti human anti-IL-1Ra antibody at 10 μg/ml or phage-display selected T111phos IL-1Ra and non-phosphorylation specific Fabs. Immunocomplexes of IL-1Ra and anti-IL-1Ra IgG were precipitated using 50 μl Protein-G (Sigma Aldrich, Prod. No. 117194116001; 2 h, 4 °C) followed by washing steps and were eluted by addition of 50 μl 0.1 M glycine (pH 3.0). Immunocomplexes of hyper/non-phosphorylated IL-1Ra with respective phospho-specific Fabs were precipitated using 20 μl Talon-beads (TaKaRa, Prod. No. 635503) for 20 min at 4 °C, followed by elution in 20 μl Imidazole at 150 mM. Subsequently western blots or isoelectric focusing was performed utilizing antibodies for IL-1Ra (antibodies online, Prod. No ABIN7123525; 1:2000), PKC (Biolegend, Prod. No. 624304; 1:2000), and PP2A (Millipore, Prod. No. 05-421; 1:2000) as described above.

## Multiplexed bead array assay of NAPKON samples

Reagents for multiplexed quantification of IL-1Ra and PGRN plasma levels in patients' samples or in samples spiked with recombinant human IL-1Ra and PGRN were purchased from R&D Systems (Minneapolis, OH, USA). All samples were prepared according to the manufacturer's instructions (R&D Systems). Data acquisition and analysis was performed on a MAGPIX instrument (Merck Millipore, Darmstadt, Germany) using xPONENT v4.2 software (Luminex).

Within NAPKON, EDTA plasma from each individual NAPKON sample was used for simultaneous profiling of 46 analytes using the Milliplex Human Cytokine/Chemokine/Growth Factor Panel A - Immunology Multiplex Assay PC 48 (Merck Millipore, Boston, MA, USA) in accordance with the manufacturer's instructions and recorded with the FlexMap3D system. The assay was performed in the affiliated NAPKON laboratory at the University Medical Center Greifswald, Greifswald, Germany.

## RNA-Seq analysis

Whole blood RNA sequencing analysis of NAPKON patients was performed at the Max-Planck Institute for Heart and Lung Research,

Bad Nauheim, Germany. Trimmomatic version 0.39 was employed to trim reads after a quality drop below a mean of Q15 in a window of 5 nucleotides and keeping only filtered reads longer than 15 nucleotides[65]. Reads were aligned versus Ensembl human genome version hg38 (Ensembl release 109) with STAR 2.7.11b[66]. Alignments were filtered to remove duplicates using Picard 3.1.1 (Picard: A set of tools (in Java) for working with next-generation sequencing data in the BAM format), multi-mapping, ribosomal, or mitochondrial reads. Gene counts were established with featureCounts 2.0.4 by aggregating reads overlapping exons on the correct strand, excluding those overlapping multiple genes[67]. The raw count matrix was normalized with DESeq2 version 1.36.0[68]. Contrasts were created with DESeq2 based on the raw count matrix. Genes were classified as significantly differentially expressed at average count > 5, multiple testing adjusted p-value < 0.05, and −0.585 < log2FC > 0.585. The Ensemble annotation was enriched with UniProt data (Activities at the Universal Protein Resource (UniProt)).

### Downstream analyses of RNA sequencing data

All downstream analyses are based on the normalized gene count matrix. Volcano and MA plots were produced to highlight DEG expression. A global clustering heatmap of samples was created based on the Euclidean distance of regularized log-transformed gene counts. Dimension reduction analyses (PCA) were performed on normalized log-transformed counts using FactoMineR[69]. Gene set overrepresentation analysis (GSO) was performed via KOBAS[70]. One test was executed per contrast using both up- and down-regulated genes to identify perturbed pathways. Bubble plots show pathways with Benjamini-Hochberg corrected p-value < 0.2 (dashed line represents pvalue < 0.05). Gray circles are scaled to the number of genes comprising the respective pathway, while smaller colored circles represent DEG subsets. Top 50 pathways were selected per contrast.

Directional gene set overrepresentation analysis (GSO) was performed via KOBAS[70]. Two separate tests were executed per contrast using only either up- or down-regulated genes to identify perturbed pathways per direction. Results were combined keeping pathways that showed significant overrepresentation at Benjamini-Hochberg corrected p-value < 0.2 (dashed line represents pvalue < 0.05) in only one input list (i.e. that were either clearly enriched for up- or down-regulated genes, but not both). Top 20 pathways were selected per contrast and direction of regulation.

Gene set enrichment analysis (GSEA) was performed via fGSEA[71]. For non-human organisms, genes were mapped to human symbols by homology based on Ensembl Biomart. All genes (not just DEGs) were ranked per contrast based on log10 transformation of the DESeq2 p-value multiplied by the negative sign of the log2 fold change. This ranks the most significantly different genes at the top/bottom. The weighting scheme for the KS statistic was set to "weighted". GSEA was performed using MSigDB collections h/c2[72]. The final table was filtered for gene sets with Benjamini-Hochberg corrected p-value < 0.2.

Transcription factor binding site (TFBS) enrichment analysis was performed with Pscan[73]. Reference TFBS position weight matrices CORE_vertebrates_non-redundant_pfms were extracted from JASPAR on 20250213[74]. Overrepresented TFBS were identified based on the promoter nucleotide sequence (450-TSS-50) of protein encoding genes using DEGs as foreground and all genes as background. The resulting heatmap shows TFs that were significantly enriched with uncorrected p-value < 0.05 in at least one foreground list (yellow = overrepresented).

### Data analysis

Data were analyzed using Graphpad Prism software (Version 8.0 for Mac OS X and version 10 for Windows, Graphpad Software, La Jolla, CA, USA). Data were analyzed for normal distribution by D'Agostino & Pearson normality test (Graphpad Prism). The large majority of data did not pass this test and were therefore subjected to non-parametric two-group analysis (Mann-Whitney test) or multi-comparison analyses by Kruskal-Wallis followed by Dunn's multiple comparison test. Comparison of proportions was done with a two-tailed Fisher's exact test. Multiple Spearman rank correlation analyses of serum analytes were performed and plotted using the corrplot R package and Rstudio (RStudio Team (2015). RStudio: Integrated Development for R. RStudio, Inc., Boston, MA http://www.rstudio.com/).

For analysis of RNA sequencing data DESeq2 DE P-values are based on Wald test that was corrected for multiple testing using the Benjamini and Hochberg method. For Kobas GSO P-values are based on hypergeometric test that was corrected for multiple testing using the Benjamini and Hochberg method. For fGSEA, P-values are based on a multilevel split Monte-Carlo scheme that was corrected for multiple testing using the Benjamini and Hochberg method. For Pscan TFBS P-values are based on z-test; no multiple testing correction was available by default. For TOBIAS, the P-value of differential binding is derived by ref. [1] fitting a two-component Gaussian mixture model to the log-transformed background score distribution to determine a threshold separating unbound and bound sites, and[2] calculating the significance threshold for the rightmost component of the distribution using a user-defined P-value. No multiple-testing correction is implemented by default.

### Reporting summary

Further information on research design is available in the Nature Portfolio Reporting Summary linked to this article.

## Data availability

All data required to evaluate the conclusions in the paper are present in the manuscript or its appendix as well as the source data file. Our study reports on proteomic and whole blood RNA sequencing data that were generated by the NAPKON consortium. Researchers can access all NAPKON data through the Use & Access procedure by submitting a formal application through the NAPKON use & access portal (https://napkon.de/use-and-access/ or https://www.netzwerk-universitaetsmedizin.de/en/research/napkon-20). All requests are reviewed by the Use and Access Committee (UAC) based on scientific merit, ethical approval from the relevant local committees, and compliance with the NAPKON Usage Regulations. For further information regarding data availability, the application process, and technical support please contact the NAPKON team via info@napkon.de. The project underlying this publication was granted access to the data set from the NAPKON UAC under the ID 2022-08-17. Source data are provided with this paper.

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

## Acknowledgements

The authors thank the CorSAAR-registry, the CAPSID trial team, the NAP-KON consortium and the German SEPNET study group for support of this study. The authors thank Sabrina Führner (Department of Pediatric Rheumatology and Immunology, University of Münster, Münster, Germany) for technical support. The authors also thank Daniela Ehlert (Department of Internal Medicine I, Saarland University, Homburg/Saar, Germany) for administrative support during the final submission process.

## Author contributions

L.T., N.F., E.R., B.T., Y.F., and C.K. planned the study. N.F., E.R., B.B., L.T., M.C.H., and C.K. performed experiments. L.T., C.K., B.T., and Y.F. wrote the manuscript. C.K. revised images and prepared the final main and Supplementary Figs. N.F., E.R., and B.B. revised the method section. C.S., F.N., I.A.K., L.T., C.K., B.T., Y.F., N.F., E.R., K.-D.P., S.H., D.K.M., T.R., K.R., O.C., S.M., A.B., F.S., C.H., C.L., J.L., A.T.-B., A.L., C.W., P.W., S.K., T.P., S.L., J.R., K.T., J.P., C.P., J.J.V., M.S., I.B., P.W., M.W., C.T., I.A., S.P., J.-P.R., M.K., V.K., K.F., R.G., H.V., D.S., S.H., U.V., S.A., S.t.H., M.D., S.B., J.B., E.D., D.P., A.P., E.H., F.B., H.S., S.S., F.L., G.G., and B.F.-O. collected and provided study samples and clinical data. F.L., G.G., and B.F.-O. contributed excellent medical assistance without which the conduct of the study would not have been possible. U.V. and S.A. provided peripheral blood multiplexed cytokine data. C.K., M.L., S.S.P., and G.A. provided RNAseq data, bioinformatic analysis thereof and drafted the respective figures, results text and methods section. L.M. and G.A. M.B., S.L.B., S.S., M.B., R.B., J.P., P.M., M.K., and P.M.L. provided study samples and clinical data, as well as administrative and organizational support. All authors approved the final version of the manuscript.

## Funding

LTh discloses support for the research of this work by a young investigator NanoBioMed grant of the University of Saarland. LTh and CKe disclose support for research and publication of this work by the Schwiete foundation. Open Access funding enabled and organized by Projekt DEAL.

## Competing interests

Statement The University of Saarland with Lorenz Thurner, Klaus-Dieter Preuss and Michael Pfreundschuh as investigators had applied for a patent on the role of PGRN as a marker for autoimmunity in 2012. The remaining authors declare no competing interests.

## Additional information

Lorenz Thurner [1] ✉, Natalie Fadle[1], Bernhard Thurner[2], Igor Age Kos[1], Moritz Bewarder[1,3], Evi Regitz[1], Birgit Bette[1], Yvan Fischer[4], Vadim Lesan[1], Torben Rixecker[5], Marie-Christin Hoffmann[1,6], Klaus-Dieter Preuss[1], Claudia Schormann[1], Dominic Kaddu-Mulindwa[1,7], Klaus Roemer[1], Onur Cetin [1], Sebastian Mang[5,8], André Becker[5], Frederik Seiler[5], Christian Herr [5], Christian Lensch[5], Johannes Lehmann[9], Angela Thiel-Bodenstaff[9,10], Andreas Link[11], Christian Werner[11], Patrick Wuchter[12], Sixten Körper[13], Thorsten Pfuhl[14], Stefan Lohse [14], Jürgen Rissland[14], Katrin Thieser[14], Jan Pilch[15], Cihan Papan[16], Sophie Roth[16], J. Janne Vehreschild [17], Margarete Scherer[17], Isabel Bröhl [17], Patricia Wagner[17], Martin Witzenrath [18], Charlotte Thibeault[18], Ira an Haack[18], Lazar Mitrov[19], Sina M. Pütz[19], Jens-Peter Reese[20,21], Michael Krawczak[22], Eckard Hamelmann[23], Verena Kopfnagel[24], Karin Fiedler[17], Ramsia Geisler[17], Heike Valentin[25], Dana Stahl[25], Sabine Hanß[26], Sabine Ameling [27], Uwe Völker [27], Stefan Hansch[28], Marcus Dörr[29], Sabine Blaschke [30], Josephine Braunsteiner[31], Edgar Dahl[32], Daniel Pape[33], Astrid Petersmann[34], Stephan Stilgenbauer [35], Frank Bloos [36], Hubert Schrezenmeier[13], Frank Langer[37], Gereon Gäbelein[38], Bettina Friesenhahn-Ochs[9], Jochen Pfeifer [39], Michael Bauer [36], Sören Leif Becker [16], Frank Neumann[1], Michael Böhm [11], Gabriele Anton [40], Carsten Kuenne[41], Soni Savai Pullamsetti[41], Mario Looso [41], Robert Bals[5], Sigrun Smola [14], Patrick Meybohm[42], Marcin Krawczyk [9,43,44,46], Philipp M. Lepper[5,10,46] & Christoph Kessel [45,46]

[1]José Carreras Center for Immuno- and Gene Therapy and Department of Internal Medicine I, Saarland University, Homburg/Saar, Germany. [2]Clinic Network Allgäu, Medical Care Center "Die Kindersprechstunde" Mindelheim, Mindelheim, Germany. [3]Department of Hematoly and Oncology, Bethanien Hospital, Frankfurt a.M, Germany. [4]Institute of Physiology, Medical Faculty, RWTH Aachen, Aachen, Germany. [5]Department of Internal Medicine V - Pulmonology, Allergology and Critical Care Medicine, Saarland University, Homburg, Germany. [6]Department of Pediatrics, University Hospital Erlangen, Erlangen, Germany. [7]Department of Oncology, Centre Hospitalier du Nord, Ettelbrück, Luxembourg. [8]Department of Intensive Care Medicine, UKE University Hospital, Hamburg, Germany. [9]Department of Medicine II, Saarland University, Homburg, Germany. [10]Department of Emergency Medicine, Saarland University, Homburg, Germany. [11]Department of Internal Medicine III–Cardiology, Saarland University, Homburg, Germany. [12]Institute of Transfusion Medicine and Immunology, Heidelberg University, Medical Faculty Mannheim, German Red Cross Blood Service of Baden-Württemberg - Hessen gGmbH, Mannheim, Germany. [13]Institute of Clinical Transfusion Medicine and Immunogenetics Ulm, German Red Cross Blood andvTransfusion Service, Baden Wuerttemberg-Hessen, and University Hospital Ulm, Ulm, Germany. [14]Institute of Virology, University of Saarland, Homburg, Germany. [15]Institute of Clinical Haemostaseology and Transfusion Medicine, Homburg, Germany. [16]Center of Infectious disease, Institute of Medical Microbiology and Hygiene, University of Saarland, Homburg, Germany. [17]Institute for Digital Medicine and Clinical Data Sciences, Faculty of Medicine, Goethe University, Frankfurt a. M, Germany. [18]Charité - Universitätsmedizin Berlin, Corporate Member of Freie Universität Berlin and Humboldt-Universität zu, Berlin, Germany. [19]Department I of Internal Medicine, Center for Integrated Oncology (CIO) Aachen Bonn Cologne Duesseldorf, University of Cologne, Medical Faculty and University Hospital Cologne, Cologne, Germany. [20]Institute for Clinical Epidemiology and Biometry, University of Würzburg, Würzburg, Germany. [21]Faculty of Health, Technische Hochschule Mittelhessen (THM), Gießen, Germany. [22]Institute of Medical Informatics and Statistics, University Medical Center Schleswig-Holstein, Kiel University, Kiel, Germany. [23]Department of Pediatrics, Children's Center Bethel, Bielefeld, University Hospital OWL, University of Bielefeld, Bielefeld, Germany. [24]Hannover Unified Biobank, Hannover Medical School, Hannover, Germany. [25]Independent Trusted Third Party of the University Medicine Greifswald, Greifswald, Germany. [26]Department of Medical Informatics, University Medical Center Göttingen, Göttingen, Germany. [27]Department of Functional Genomics, Interfaculty Institute for Genetics and Functional Genomics, University Medicine Greifswald, Greifswald, Germany. [28]Department for Infectious Diseases and Infection Control, University Hospital Regensburg, Regensburg, Germany. [29]Department of Internal Medicine B, University Medicine Greifswald, Greifswald, Germany. [30]Emergency Department, University Medical Center, Goettingen, Germany. [31]Center for Anesthesiology and Intensive Care Medicine, University Hospital UKE, Hamburg-Eppendorf, Hamburg, Germany. [32]RWTH centralized Biomaterial Bank (RWTH cBMB), Institute of Pathology, Medical Faculty, RWTH Aachen University, Aachen, Germany. [33]Department I of Internal Medicine, University Medical Center Schleswig-Holstein Campus Kiel, Kiel, Germany. [34]Institute for Clinical Chemistry and Laboratory Medicine, University Medicine Oldenburg, Oldenburg, Germany. [35]Comprehensive Cancer Center Ulm and Division of CLL-Internal Medicine III, Ulm University, Ulm, Germany. [36]Department of Anesthesiology and Intensive Care Medicine, Jena University Hospital, Jena, Germany. [37]Department of Thoracic Surgery, Saarland University Medical Center, Homburg/Saar, Germany. [38]Department of General, Visceral, Vascular and Pediatric Surgery, University of Saarland, Homburg, Germany. [39]Department of Pediatric Cardiology, University of Saarland,

Homburg, Germany. [40]Medical School East Westphalia-Lippe, Bielefeld University, Bielefeld, Germany. [41]Max-Planck Institute for Heart and Lung Research, Bad Nauheim, Germany. [42]Department of Anesthesiology, Intensive Care, Emergency and Pain Medicine, University Hospital Würzburg, Würzburg, Germany. [43]Department of Gastroenterology, Hepatology and Transplant Medicine, Medical Faculty, University of Duisburg-Essen, Essen, Germany. [44]Laboratory of Metabolic Liver Diseases, Centre for Preclinical Research, Department of General, Transplant and Liver Surgery, Medical University of Warsaw, Warsaw, Poland. [45]Department of Pediatric Rheumatology and Immunology, University of Münster, Münster, Germany. [46]These authors contributed equally: Marcin Krawczyk, Philipp M. Lepper, Christoph Kessel. ✉e-mail: lorenz.thurner@uks.eu

