## [Transparent Peer Review file · Nature Communications]

Autoantibodies to IL-1Ra and PGRN in severe COVID-19 are associated with inflammation-induced hyperphosphorylated antigen isoforms

Corresponding Author: Professor Lorenz Thurner

Version 0:

Reviewer comments:

Reviewer #1

(Remarks to the Author)

Authors have made an interesting observation that autoantibodies to IL-1Ra and PGRN in severe COVID-19 are associated with inflammation-induced hyperphosphorylated antigen isoforms. Importantly, such patients did not show autoantibodies to type I IFN reported earlier.

1. Analysis of antibodies from severe inflammatory cases (infectious or noninfectious) is necessary to determine if the observations are specific to COVID-19 or not
2. Functional characterization of autoantibodies by neutralization assays and SPR is needed.
3. By what mechanisms IL-1RA and PGRN undergo hyperphosphorylation ?
4. Whether Adalimumab or Infliximab show impaired anti-TNF action in these patients? In vitro assays can be performed.

Reviewer #2

(Remarks to the Author)

The manuscript by Thurner et al. presents research on autoantibody responses against IL-1Ra and PGRN and their relation with hyperinflammation in severe COVID-19. The authors found that hyperphosphorylated antigen forms and autoantibodies against IL-1Ra and PGRN are found and coincide in severe COVID-19. Moreover, they show that these autoantibodies interfere with the anti-inflammatory function of the antigen and could relate seropositivity with markers for inflammation. Thereby, the manuscript provides a potential link between an autoimmune response and inflammation in severe COVID-19.

Comments

We consider the manuscript potentially novel and relevant. It extends the already established knowledge on autoantibody responses observed in COVID-19 and provides a link with hyperinflammation. The experimental approaches used by the authors seem logical and solid. Experiments are mostly well-controlled and complementary approaches are used.

Nevertheless, we have multiple issues that require further clarification:

- The authors conclude from figure 1B and 1D that PGRN/IL-1Ra-IgM/G immune complexes are present in the circulation of seropositive patients. The detection of immune complexes in serum can be considered a surprising finding, given, depending on their size, the fast clearance of immune complexes in the circulation. More importantly, the observed apparent molecular weights do not seem to correspond with theoretical masses (E.g. PGRN – is indicated at 14.2kDa; estimated size ~70kDa, PGRN+IgG complex – is indicated at 132kDa instead of estimated ~150 (size Igg)+70kDa, PGRN+IgM complex – is indicated at 660kDa instead of estimated ~>900kDa+70kDa, same applies for IL-1Ra). These apparent inconsistencies raise serious concerns to the interpretation of the data and should be clarified.
- Likewise, these data imply the sole presence of immune complexes of one antibody with one antigen which would indicate that the antibodies only bind to one epitope of the antigen. Similarly, it indicates that one antibody does not bind with both arms to antigen. Based on these considerations, doubt is raised on the interpretations of the authors to these results, and we

request clarification on the interpretation by the authors and on the methodology used to generate figures 1B and 1D of the data in figure 1B and figure 1D (e.g. what is exactly depicted in the these figures).

- If antigen-antibody complexes are present in the circulation, this could interfere with the measurements of antibodies, antigen, and phosphorylated antigen. Nevertheless, the possibility of such interference is not thoroughly discussed in the manuscript.
- Why do the authors consider that hyperphosphorylation of IL-1Ra and PGRN is only observed in seropositive patients with severe COVID-19? The authors show that various cytokines can induce hyperphosphorylation of IL-1Ra and PGRN in monocytes obtained from convalescent COVID-19 patient and healthy controls (albeit at higher concentrations for healthy donors). Moreover, the authors identify the enzymes responsible for the hyperphosphorylation of IL-1Ra and PGRN. Yet, hyperphosphorylation of IL-1Ra and PGRN is only found in seropositive patients. However, hyperinflammation and cytokines that can induce hyperphosphorylation are presumably also present in seronegative patients with severe COVID-19 (and sepsis patients with a 'pulmonary' focus which were used as controls). So, why do the authors think that hyperphosphorylation of IL-1Ra and PGRN is specifically detected in seropositive patients? Is hyperphosphorylation differentially regulated in seropositive patients and is this inherent to these patients? And do the authors expect that hyperphosphorylation of IL-1Ra and PGRN is also to be present in other situations, e.g. other respiratory viruses?
- It is unclear whether the authors expect a direct link between hyperphosphorylation of antigen and the break in B cell tolerance against these antigens. Insights into this aspect, including a potential role of T cells in provision of helper activity to these B cells, in the discussion section would be helpful to guide the reader the potential mechanisms underlying the observations presented.

In conclusion, we consider the manuscript potentially novel and relevant. However, in our opinion, the points raised above have to be addressed to approve the manuscript for publication.

Reviewer #3

(Remarks to the Author)

This is an important paper that demonstrates the presence of autoantibodies against IL-1Ra and PGRN in patients with severe COVID-19. Since IL-1Ra and PGRN are involved in suppressing the inflammatory response, these autoantibodies may contribute to the pathogenesis of severe COVID-19. The study is well organized, and the results are clear. However, there are a few points that could enhance the clarity of the paper:

1. The data presented in Figure 2E are unclear. Do they depict autoantibody levels specific to phosphorylated IL-1Ra or PGRN, or do they merely show the levels of the phosphorylated proteins in patients using antibodies specific to the phosphorylated forms? A more detailed description in the figure legend would be helpful. Moreover, since monoclonal antibodies specific to the phosphorylation sites are available, a competitive binding assay could potentially be employed to detect autoantibodies that target those sites.
2. What is the pathogenic function of the autoantibodies against IL-1Ra and PGRN? Although the authors demonstrated that serum concentrations of IL-1Ra and PGRN increased as the autoantibody levels decreased, could these autoantibodies interfere with the biological functions of IL-1Ra and PGRN? It may be worth exploring the functional activity of these proteins in the presence of patients' serum autoantibodies.
3. Why do monocytes from severe COVID-19 patients tend to produce phosphorylated IL-1Ra and PGRN? A comparative analysis of monocytes from healthy controls and severe COVID-19 patients using scRNA-seq might elucidate the underlying mechanisms. Public scRNA-seq datasets of severe COVID-19 patients could potentially be analyzed to predict potential mechanisms.

Version 1:

Reviewer comments:

Reviewer #1

(Remarks to the Author)

The authors have satisfactorily addressed the concerns of this reviewer

Reviewer #2

(Remarks to the Author)

The authors addressed most comments adequately. However, the answers given to the concern raised around the molecular masses of the proteins detected in the experiments depicted in figure 1 and new figure S7 need further clarification.

- The molecular masses of the proteins depicted in figure 1B/D, differ from the molecular masses depicted in the new figure S7D. The latter are in line with immune complexed antigens, the data presented in figure 1B/d do not seem to be in line with this notion. This should be clarified.

- In figure S7D a band named P4 is depicted. No peak 4 is present in the data presented in figure S7C.

- In figure S7D/E, a band around 66 Kd is present. No such band is present in figure 1 of the main manuscript.

- No negative controls seem to be included in the new dataset depicted in figure S7.

Reviewer #3

(Remarks to the Author)

The authors revised the manuscript properly. Nothing to revise anymore.

Version 2:

Reviewer comments:

Reviewer #2

(Remarks to the Author)

I do not have additional comments or questions.

Dear Editors, Dear Reviewers,

Thank you very much for your very constructive feedback.

We have addressed all raised points in this point-by-point reply letter.

All changes according to reviewers' comments are highlighted in **red font**. Additional edits introduced by us to maintain text flow or correct mistakes we have identified outside of reviewer comments are highlighted in **blue font**.

The respective page and line count in this point-by-point reply refers to the track-changes version of the revised manuscript.

Kind regards,

Lorenz Thurner and Christoph Kessel, on behalf of all study authors

Reviewer 1

Authors have made an interesting observation that autoantibodies to IL-1Ra and PGRN in severe COVID-19 are associated with inflammation-induced hyperphosphorylated antigen isoforms. Importantly, such patients did not show autoantibodies to type I IFN reported earlier.

Comment	Authors' reply	New/revised text/item (changes in red font)		
		page/item	line	
MAJOR COMMENTS				
1. Analysis of antibodies from severe inflammatory cases (infectious or noninfectious) is necessary to determine if the observations are specific to COVID-19 or not	Thank you, we absolutely agree with this interpretation. To already address this point in course of the original drafting of the manuscript, we included samples of two multi-center pre-pandemic prospective trials of the German SEPNET with patients with sepsis and pulmonary focus in this study for post-hoc analysis (SISPCT, NCT00832039; MaxSep, NCT00534287, n=174; Figure 1F). As stated in the original manuscript's results section, anti-PGRN and anti-IL-1Ra antibodies were detected only infrequently in those patients (anti-PGRN: 4/174; anti-IL-1Ra: 3/174). In the revised manuscripts results text, we indicated the respective trial numbers. Further, we want to point out, that following initial publication of several of the data also included in the present manuscript as pre-print (doi.org/10.1101/2021.04.23.441188), we completed two studies also	Results, page 8 Discussion, page 19	26-33 33 - ...	In contrast, no anti-PGRN and anti-IL-1Ra antibodies were detected in several control cohorts including healthy pre-pandemic controls (n=188), healthcare-workers following 2 vaccinations against SARS-CoV2 (n=40), and SARS-CoV2 exposed children and adolescents enrolled in the COKIBA registry (n=146). Anti-PGRN and anti-IL-1Ra antibodies were detected infrequently in patients with a pulmonary focus included in two prepandemic multi-center sepsis trials (SISPCT; NCT00832039 (n=121): anti-PGRN: 3/121 (2.5%); anti-IL-1Ra: 2/121 (1.7%); and MaxSep; NCT00534287 (n=53): anti-PGRN: 1/53 (1.9%); anti-IL-1Ra: 1/53 (1.9%)) (Fig. S1 and Fig. 1F). Altogether and based on the present data we currently hypothesize that inflammation itself can drive transient hyperphosphorylation of IL-1Ra and PGRN, which may then result in a break of peripheral tolerance on B and T cell level depending on a genetic predisposition of individuals which remains yet to be defined. Furthermore, our observations on impact of inflammatory stress (i.e. mediated by signaling of prominent inflammatory cytokines) and induction of

	reporting anti-IL-1Ra antibodies outside of a SARS-CoV2-context (Still's Disease (Hoffmann et al., J Clin Immunol, 2024); recurrent idiopathic pericarditis (Wu et al., JAMA Netw Open, 2025)). Beyond, anti-IL-1Ra antibodies have been reported from IgG-4 related disease (Jarrell et al., JACI, 2022; referenced in the original manuscript) and in context of pre-term pregnancy (MedRxiv: doi.org/10.1101/2024.10.03.24314850). With respect to anti-PGRN antibodies and as referenced in the original manuscript, those have already been reported in several inflammatory conditions (i.e. Thurner et al., J Autoimmune, 2013) prior to the SARS-CoV-2 pandemic. Based on all those findings, our present interpretation is, that anti-PGRN and anti-IL-1Ra antibodies are not specific for a COVID-19 context, but rather associate with severe inflammation. We have included a respective statement in the revised manuscript's discussion.		hyperphosphorylation suggest that this and related autoantibody responses may not be limited to a SARS-CoV-2 infection and linked hyperinflammatory context. This is supported by data from us and others reporting on anti-IL-1Ra and/or anti-PGRN antibodies in inflammatory conditions not associated with SARS-CoV-2 (18, 21, 34, 56, 57).
--	---	--	---

			tagged SARS-CoV-2 S1-, S2-, E- or M-proteins (Fig. S5A, B). In addition, antibodies against SARS-CoV-2 S1-, S2-, or M-proteins as well as against human IL-1Ra could not be adsorbed by immobilized PGRN or IL-1Ra, but were detectable in the eluate of samples from patients with severe or critical COVID-19, excluding cross-reactivity of anti-PGRN and anti-IL-1Ra (Fig. S5A, B). To further confirm specificity of the identified autoantibodies, we performed competition experiments for binding to plate-bound antigen of IgG purified from patients' plasma vs. commercial anti-IL-1Ra (Fig. S5C, D) or anti-PGRN mAbs (Fig. S5E, F). These experiments demonstrated that purified IgG could outcompete commercial mAb binding in a concentration dependent manner and vice versa (Fig. S5C-F).
--	--	--	--

3. By what mechanisms IL-1RA and PGRN undergo hyperphosphorylation?

For IL-1Ra, we identified protein kinase C (PKC) as the responsible kinase, and protein phosphatase 1 (PP1) as the counteracting phosphatase. Experiments already described in the original manuscript involved incubation of patient- and control-derived primary monocytes with kinase/phosphatase inhibitor panels. These findings were independently confirmed by Co-IP (original manuscript **Figure S9**; **Figure S11** in the revised manuscript).

For PGRN, we previously demonstrated that Ser81 hyperphosphorylation is mediated by PKC β 1 and reversed by PP1, using analogous inhibitor panel experiments in patient- and control-derived lymphoblastoid cell lines (LCLs) (Turner et al., *J Autoimmune*, 2015). These results were likewise validated by Co-IP. Notably, in PGRN autoantibody-positive, pSer81-positive patients, an inactivated PP1 isoform was found to be specifically complexed with PGRN, suggesting impaired dephosphorylation in this subgroup.

One possible reason for the observed transient appearance of hyperphosphorylated isoforms may be an inflammatory environment-mediated induction of this modification. Both LCLs and primary monocytes exposed to inflammatory cytokines reproducibly developed

	hyperphosphorylated forms of PGRN and IL-1Ra. Strikingly, monocytes from survivors of critical COVID-19 required markedly lower cytokine thresholds - even one year after infection - to generate these isoforms, indicating a sustained predisposition to hyperphosphorylation (original manuscript Figure 4H, I). These findings may suggest a two-hit model of disease, in which an initial inflammatory milieu triggers abnormal phosphorylation, thereby creating a substrate for autoantibody generation and perpetuating autoimmune-mediated hyperinflammation.			
--	--	--	--	--

4. Whether Adalimumab or Infliximab show impaired anti-TNF action in these patients? In vitro assays can be performed.	Thank you for this suggestion. Accordingly, we tested both adalimumab and etanercept in an MTT assays using WEHI-S cells in concentrations of 0.15 µg/ml up to 2 µg/ml in the presence of diluted patient plasma with or without PGRN antibodies. In this context we revised Figure S7 (now Figure S8) accordingly and included respective results text in the revised manuscript.	Figure S8, C, D Results, page 10	7 - 18	When analyzed by commercial ELISAs PGRN levels were decreased by more than 90% in the antibody-positive patients (mean: 21.8 ng/mL), compared to (i) plasma of anti-PGRN-antibody-negative patients (mean: 245.8 ng/mL) and (ii) plasma of seronegative patients hospitalized in ICU outside COVID19 context (mean 212.2 ng/mL) (Fig. S8A). Similarly, IL-1Ra plasma levels were decreased by 78% in anti-IL-1Ra-antibody-positive patients (mean: 258.0 pg/ml), compared to plasma of seronegative patients (mean 1933 pg/ml) (Fig. S8E). Anti-PGRN and anti-IL-1RA antibodies in patients' plasma also resulted in a functional impairment of PGRN and IL-1Ra bioactivity, as determined by TNF- and IL-1-signaling reporter cell lines, respectively (Fig. S8B-D, F, G). In these assays, clinically approved drugs such as Anakinra (recombinant human IL-1Ra), Adalimumab (anti-TNF mAb) or Ethanercept (TNFR1-IgG Fc fusion protein) could reverse the impact of anti-PGRN (Fig. S8C, D) or anti-IL-1Ra autoantibodies (Fig. S8F, G) on IL-1β- or TNF-signaling, respectively.
---	--	--	-------------------	---

Reviewer 2

The manuscript by Thurner et al. presents research on autoantibody responses against IL-1Ra and PGRN and their relation with hyperinflammation in severe COVID-19. The authors found that hyperphosphorylated antigen forms and autoantibodies against IL-1Ra and PGRN are found and coincide in severe COVID-19. Moreover, they show that these autoantibodies interfere with the anti-inflammatory function of the antigen and could relate seropositivity with markers for inflammation. Thereby, the manuscript provides a potential link between an autoimmune response and inflammation in severe COVID-19.

Comments

We consider the manuscript potentially novel and relevant. It extends the already established knowledge on autoantibody responses observed in COVID-19 and provides a link with hyperinflammation. The experimental approaches used by the authors seem logical and solid. Experiments are mostly well-controlled and complementary approaches are used. Nevertheless, we have multiple issues that require further clarification:

Comment	Authors' reply	New/ revised text/item (changes in red font)		
		page/ item	line	
The authors conclude from figure 1B and 1D that PGRN/IL-1Ra-IgM/G immune complexes are present in the circulation of seropositive patients. The detection of immune complexes in serum can be considered a surprising finding, given, depending on their size, the fast clearance of immune complexes in the circulation. More importantly, the observed apparent molecular weights do not seem to correspond with theoretical masses (E.g. PGRN – is indicated at 14.2kDa; estimated size ~70kDa, PGRN+IgG complex – is indicated at 132kDa instead of estimated ~150 (size Igg)+70kDa, PGRN+IgM complex – is indicated at 660kDa instead of estimated ~>900kDa+70kDa, same applies for IL-1Ra). These apparent inconsistencies raise serious concerns to the interpretation of the data and should be clarified.	Thank you, this is a very valid point. It is of course correct that immune complexes can be efficiently cleared from the circulation by FcR expressing APCs. However, from autoimmune conditions such as SLE, RA or vasculitis a high prevalence of circulating immune complexes has been reported (i.e. Tung et al., Clin Exp Immunol. 1981; Zhao et al., Arth Res Ther. 2008; Matsumoto et al., Sci Adv. 2022). Similarly, in COVID-context colleagues from Freiburg (Germany) whom we already previously discussed our data with demonstrated a high prevalence of circulating multimeric immune complexes in COVID-19 (Ankerhold et al., Nat Com. 2022).	new Figure S7 Results, page 9	26 - ...	To further characterize the observed immune complexes in patients' plasma, we performed size-exclusion chromatography in order to separate non-antibody bound and immune-complexed antigens (Fig. S7A-D). In case of immune complexed IL-1Ra, we observed three major peaks corresponding to non-immune complexed protein (~20 kDa), IgG- (~177 kDa), and IgM-complexed (~870 kDa) IL-1Ra (Fig. S7B), which we validated by immunoblotting (Fig. S7D). Similarly, PGRN eluted as unbound protein (~15kDa: granulin, ~65 kDa: progranulin), IgG- (~202 kDa), and IgM-complexed (~903 kDa) PGRN (Fig. S7C), consistent with the presence of both low- and high-molecular-weight immune complexes in patient plasma (Fig. S7D). Furthermore, in order to verify respective sizes of immune complexes detected in patients' plasma,

	Such observations may argue in favor of detecting such immune complexes in serum or plasma of patients. Beyond – and using the same methodology as outline in our previous studies and the present manuscript – we also just recently validated the presence of anti-OPG antibodies and respective immune complexes in axial spondyloarthritis (Annals of the Rheum Dis, revision invited), as originally reported elsewhere (Riches et al., NEJM, 2009; Hauser et al., Calcif Tissue Int, 2017). In order to address the raised issues regarding immune complex size we performed additional experiments, including formation of artificial immune complexes and FPLC analysis of immune complexes in patients' plasma. Respective experiments and data are outlined in the revised manuscript (please see new Figure S7 and related methods and results text).			we generated artificial immune complexes using recombinant antigen and mono- or polyclonal commercial IL-1Ra or PGRN-specific IgG (Fig. S7E, F). These experiments confirmed the banding patterns of IgG-associated species as observed with immune complexes detected in patients' plasma according to native gradient PAGE or native PAGE at fixed resolution and subsequent Western blotting (Fig. S7E-G).
Likewise, these data imply the sole presence of immune complexes of one antibody with one antigen which would indicate that the antibodies only bind to one epitope of the antigen. Similarly, it indicates that one antibody does not bind with both arms to antigen. Based on these considerations, doubt is raised on the interpretations of the authors to these results, and we request clarification on the interpretation by the authors and on the methodology used to generate figures 1B and 1D of the data in	Thank you for triggering this discussion! In fact, our new data on artificial immune complexes using mono- and polyclonal antibodies (please see new Figure S7 and related results text) would support an interpretation of a 1:1 antigen:antibody ratio for both IL-1Ra and PGRN. It is of note here, that our previously reported rough epitope mapping approach using recombinant IL-1Ra	new Figure S7 Results, page 9	26 - ...	To further characterize the observed immune complexes in patients' plasma, we performed size-exclusion chromatography in order to separate non-antibody bound and immune-complexed antigens (Fig. S7A-D). In case of immune complexed IL-1Ra, we observed three major peaks corresponding to non-immune complexed protein (~20 kDa), IgG- (~177 kDa),

figure 1B and figure 1D (e.g. what is exactly depicted in the these figures).	fragments already indicated that (in contrast to MIS-C or Still's Disease) anti-IL-1Ra antibodies in context of severe COVID-19 can recognize several antigenic determinants on IL-1Ra (Hoffmann et al., J Clin Immunol, 2024), two of which overlap with epitopes identified for anti-IL-1Ra antibodies observed in context of IgG4-related disease (Jarrell et al., JACI, 2022). Strikingly, a recent pre-print from Joe DeRisi's lab (doi.org/10.1101/2024.10.03.24314850) indicated, that for anti-IL-1Ra antibodies detected in context of pre-term pregnancies only one epitope-specificity was determined, arguing for a very focused response. Importantly, this epitope also overlaps with one identified by us as well as Jarrell et al (please see reply letter Figure 1). With respect to data as presented in Fig 1B and D, those show Western Blot of non-reducing gradient PAGE of plasma of severe COVID-19 patients followed by Western blotting using commercial anti-human PGRN and commercial anti-human IL-1Ra antibodies for detection, respectively. Corresponding to ELISA data depicted in Fig. 1A seropositive patients' plasma reveals in part reduced free PGRN/IL-1Ra levels (less band intensity in WB), while	Figure,1 caption	and IgM-complexed (~870 kDa) IL-1Ra (Fig. S7B), which we validated by immunoblotting (Fig. S7D). Similarly, PGRN eluted as unbound protein (~15kDa: granulin, ~65 kDa: progranulin), IgG (~202 kDa), and IgM-complexed (~903 kDa) PGRN (Fig. S7C), consistent with the presence of both low- and high-molecular-weight immune complexes in patient plasma (Fig. S7D). Furthermore, in order to verify respective sizes of immune complexes detected in patients' plasma, we generated artificial immune complexes using recombinant antigen and mono- or polyclonal commercial IL-1Ra or PGRN-specific IgG (Fig. S7E, F). These experiments confirmed the banding patterns of IgG-associated species as observed with immune complexes detected in patients' plasma according to native gradient PAGE or native PAGE at fixed resolution and subsequent Western blotting (Fig. S7E-G). Next to ELISA, severe and critical COVID19 discovery cohort plasma samples (n=30) were also assessed for presence of (B) anti-PGRN and (D) anti-IL-1Ra IgM and IgG immune complexes using non-reducing native PAGE followed by western blot. (C, E) Further, severe and critical COVID19 discovery cohort plasma samples (n=30) were tested by IEF for occurrence of (C) PGRN and (E) IL-1Ra isoforms."
--	---	------------------------------------	--

	immune complexed PGRN/IL-1Ra is detectable.			
If antigen-antibody complexes are present in the circulation, this could interfere with the measurements of antibodies, antigen, and phosphorylated antigen. Nevertheless, the possibility of such interference is not thoroughly discussed in the manuscript.	It is of course correct that endogenous immune complexes may sequester the free analyte (antigen or antibody) and may alter binding kinetics in immune assay systems. This can result in falsely low measured concentrations or distort assay read-outs. Immune complexes may mask epitopes required for capture/detection reagent binding or may bridge assay antibodies and thus lead to under- or over-estimation of analyte concentration (Tate and Ward, Clin Biochem Rev , 2004). We do hope that this is what you were referring to and drafted some revised discussion in these lines.	Discussion , page 17	6 - 13	In context of all immune assay formats applied by us and others it requires to be stressed that endogenous immune complexes may sequester the free analyte (antigen or antibody) and can alter binding kinetics in such assay systems. Immune complexes may mask epitopes required for capture and/or detection reagent binding or may bridge assay antibodies and thus lead to under- or over-estimation of analyte concentration (47). In this respect, we consider our approach of combining a biochemical assessment of immune-complexed and unbound antigen using native gradient gels and Western blotting with immune assays (ELISA) a strength of our work.
Why do the authors consider that hyperphosphorylation of IL-1Ra and PGRN is only observed in seropositive patients with severe COVID-19? The authors show that various cytokines can induce hyperphosphorylation of IL-1Ra and PGRN in monocytes obtained from convalescent COVID-19 patient and healthy controls (albeit at higher concentrations for healthy donors). Moreover, the authors identify the enzymes responsible for the hyperphosphorylation of IL-1Ra and PGRN. Yet, hyperphosphorylation of IL-1Ra and PGRN is only found in seropositive patients. However, hyperinflammation and cytokines that can induce hyperphosphorylation are presumably also present in seronegative patients with severe COVID-19 (and sepsis patients with a	Thanks, this is of course highly relevant in context of a broader scope of the presented findings. Indeed, our previous (Hoffmann et al., J Clin Immunol , 2024) but also present work (Wu et al., JAMA Netw Open , 2025) outside of SARS-CoV context demonstrate hyperphosphorylation of particularly IL-1Ra also to occur in other inflammatory scenarios. Similarly, previous work in context of autoimmune conditions reported similar findings for PGRN (Thurner et al., J Autoimmune , 2015). Of all our investigations in this respect, the present study is the so	Discussion , page 19	33 - ...	Altogether and based on the present data we currently hypothesize that inflammation itself can drive transient hyperphosphorylation of IL-1Ra and PGRN, which may then result in a break of peripheral tolerance on B and T cell level depending on a genetic predisposition of individuals which remains yet to be defined. Furthermore, our observations on impact of inflammatory stress (i.e. mediated by signaling of prominent inflammatory cytokines) and induction of hyperphosphorylation suggest that this and related autoantibody responses may not be limited to a SARS-CoV-2 infection and linked hyperinflammatory context. This is supported by data from us and others reporting on anti-IL-1Ra and/or anti-PGRN antibodies in inflammatory conditions not associated with SARS-CoV-2 (18,

'pulmonary' focus which were used as controls). So, why do the authors think that hyperphosphorylation of IL-1Ra and PGRN is specifically detected in seropositive patients? Is hyperphosphorylation differentially regulated in seropositive patients and is this inherent to these patients? And do the authors expect that hyperphosphorylation of IL-1Ra and PGRN is also to be present in other situations, e.g. other respiratory viruses?	far largest, which also offered the opportunity for the reported cellular analysis. In a follow-up to the PERIPLO study (Wu et al., JAMA Netw Open, 2025) we are also planning for this, but this is yet to come. From all our data collected so far, we conclude that inflammation itself can be a driver of hyperphosphorylation. In genetically susceptible individuals this can result in a transient peripheral break of tolerance and respective autoantibody responses. In terms of underlying genetics we are currently initializing analysis of >3000 data sets from the NAPKON registry in Germany and link those with serological autoantibody status. Yet, this is still in a negotiation phase. In the revised manuscript we have included additional discussion regarding our view on hyperphosphorylation and link with COVID-19 and other (hyper)inflammatory conditions.			21, 34, 56, 57).
It is unclear whether the authors expect a direct link between hyperphosphorylation of antigen and the break in B cell tolerance against these antigens. Insights into this aspect, including a potential role of T cells in provision of helper activity to these B cells, in the discussion section would be helpful to	The present manuscript already contains detailed discussion on T and B cells and how we envision those to be involved in the reported observations (please see text in the right column). We have included a summarizing statement in the	Discussion, page 17	25 - ...	On an anecdotal note, one patient enrolled in the discovery cohort was tested for anti-PGRN and anti-IL-1Ra antibodies within a 7-day interval before and after one cycle of plasmapheresis. This resulted in a rapid drop of autoantibody titers (Fig. S15A-C), and paralleling recovery of IL-1Ra plasma levels (Fig. S15D). In course of

guide the reader the potential mechanisms underlying the observations presented.	discussion section of the revised manuscript.	Discussion, page 19 Discussion, page 19	14- ... 33 - ...	plasmapheresis hyperphosphorylated IL-1Ra disappeared, while the hyperphosphorylated isoform of PGRN remained detectable (Fig. S15E). Even though we have not yet formerly demonstrated, that hyperphosphorylated antigen can break peripheral tolerance, these data may further strengthen such causal link. In other context reported previously (38, 44-46) we already managed to identify phospho-antigen specific T cell responses (47) while phosphorylated T cell epitopes have been assigned high MHCII affinity and to induce strong CD4 and CD8 T cell responses (48, 49). In case of anti-PGRN and anti-IL-1Ra antibodies as described in the present study, the transience of the antibody response may also argue for the involvement of an extra-follicular B cell activation, resulting in formation of rather short-lived antibody producing plasma cells and no establishment of B cellular memory. Extrafollicular B cell responses can be driven and supported by inflammatory stress (50, 51). Our data indicate Collectively, in severe COVID-19 our data point at an association of anti-IL-1Ra and anti-PGRN antibodies with load of inflammation... Altogether and based on the present data we currently hypothesize that inflammation itself can drive transient hyperphosphorylation of IL-1Ra and PGRN, which may then result in a break of peripheral tolerance on B and T cell level depending on a genetic predisposition of individuals which remains yet to be defined. Furthermore, our observations on impact of inflammatory stress (i.e. mediated by signaling of
---	--	---	--------------------------------	--

				prominent inflammatory cytokines) and induction of hyperphosphorylation suggest that this and related autoantibody responses may not be limited to a SARS-CoV-2 infection and linked hyperinflammatory context. This is supported by data from us and others reporting on anti-IL-1Ra and/or anti-PGRN antibodies in inflammatory conditions not associated with SARS-CoV-2 (18, 21, 34, 56, 57).
In conclusion, we consider the manuscript potentially novel and relevant. However, in our opinion, the points raised above have to be addressed to approve the manuscript for publication.	Thank you for your time dedicated for assessing our manuscript and the raised important discussion in context of immune complexes. Throughout, we hope that we understood all questions correctly and adequately replied to those.			

Reviewer 4

This is an important paper that demonstrates the presence of autoantibodies against IL-1Ra and PGRN in patients with severe COVID-19. Since IL-1Ra and PGRN are involved in suppressing the inflammatory response, these autoantibodies may contribute to the pathogenesis of severe COVID-19. The study is well organized, and the results are clear. However, there are a few points that could enhance the clarity of the paper:

Comment	Authors' reply	New/revised text/item (changes in red font)		
		page/ item	line	
MAJOR COMMENTS				
1. The data presented in Figure 2E are unclear. Do they depict autoantibody levels specific to phosphorylated IL-1Ra or PGRN, or do they merely show the levels of the phosphorylated proteins in patients using antibodies specific to the phosphorylated forms? A more detailed description in the figure legend would be helpful. Moreover, since monoclonal antibodies specific to the phosphorylation sites are available, a competitive binding assay could potentially be employed to detect autoantibodies that target those sites.	Sorry for not being clear here. Figure 2E shows data for recognition of recombinant IL-1Ra or PGRN by IgG in patients' plasma side-by-side with presence of hyperphosphorylated IL-1Ra or PGRN in those plasma samples. These data link seropositivity for either anti-IL-1Ra or anti-PGRN antibodies with presence of hyperphosphorylated antigen in patients' plasma samples. We have included a respective statement in the Fig. 2 caption. Beyond, we now performed competition experiments of plasma IgG vs Fabs specific for either canonically or hyperphosphorylated IL-1Ra. Data are included as new Figure S11 and described in the revised manuscript's results section.	Fig. 2 caption new Figure S10 Results, page 11 Discussion, page 17	20 - 23 22 - 24	Data show recognition of recombinant PGRN or IL-1Ra by IgG in patients' plasma (dark blue or dark red bars) side-by-side with presence of hyperphosphorylated PGRN or IL-1Ra in those plasma samples (light blue/light red bars). When using wt-IL-1Ra or T111^{phos} IL-1Ra-specific Fabs in competition experiments with ex vivo IgG, we observed antibodies in patients' plasma to compete with both wt-IL-1Ra- as well as T111^{phos} IL-1Ra-specific Fabs for binding to plate-bound canonically phosphorylated or hyperphosphorylated IL-1Ra, respectively (Fig. S10). Importantly, competition experiments of selected Fabs against patients' IgG revealed, that plasma antibodies against IL-1Ra or PGRN target both a

				phosphorylated epitope, but also unphosphorylated antigenic determinants.
2. What is the pathogenic function of the autoantibodies against IL-1Ra and PGRN? Although the authors demonstrated that serum concentrations of IL-1Ra and PGRN increased as the autoantibody levels decreased, could these autoantibodies interfere with the biological functions of IL-1Ra and PGRN? It may be worth exploring the functional activity of these proteins in the presence of patients' serum autoantibodies.	Thank you, this is of course an important point. Already in previous work we demonstrated that both anti-IL-1Ra as well as anti-PGRN antibodies not only deplete their respective antigens from the circulation, but also impair their bioactivity. This we assessed by using reporter cell lines for either IL-1β- or TNF-signaling (Thurner et al., J Autoimmune, 2015; Pfeifer et al., Lancet Rheum, 2022; Thurner et al., NEJM, 2022; Hoffmann et al., J Clin Immunol, 2024) but also present work (Wu et al., JAMA Netw Open, 2025). Consequently, we also tested this in context of the present study (original manuscript Figure S7, now Figure S8), which – upon request by reviewer 1 - has also been expanded by demonstrating that therapeutic TNF-blocking agents (ethanercept, adalimumab) can override the PGRN-neutralizing effect of anti-PGRN autoantibodies. In the lines of your comment as well as additional experiments suggested by reviewer 1, we have revised the manuscript accordingly.	Figure S8 Results , page 10	7 - 18	When analyzed by commercial ELISAs PGRN levels were decreased by more than 90% in the antibody-positive patients (mean: 21.8 ng/mL), compared to (i) plasma of anti-PGRN-antibody-negative patients (mean: 245.8 ng/mL) and (ii) plasma of seronegative patients hospitalized in ICU outside COVID19 context (mean 212.2 ng/mL) (Fig. S8A). Similarly, IL-1Ra plasma levels were decreased by 78% in anti-IL-1Ra-antibody-positive patients (mean: 258.0 pg/ml), compared to plasma of seronegative patients (mean 1933 pg/ml) (Fig. S8E). Anti-PGRN and anti-IL-1RA antibodies in patients' plasma also resulted in a functional impairment of PGRN and IL-1Ra bioactivity, as determined by TNF- and IL-1-signaling reporter cell lines, respectively (Fig. S8B-D, F, G). In these assays, clinically approved drugs such as Anakinra (recombinant human IL-1Ra), Adalimumab (anti-TNF mAb) or Ethanercept (TNFR1-IgG Fc fusion protein) could reverse the impact of anti-PGRN (Fig. S8C, D) or anti-IL-1Ra autoantibodies (Fig. S8F, G) on IL-1 β or TNF-signaling, respectively.
3. Why do monocytes from severe COVID-19 patients tend to produce phosphorylated IL-1Ra and PGRN? A comparative analysis of	Thank you, this is indeed a highly intriguing question.	new Figure S13		

| | | | | |
|---|---|---|---|---|
| monocytes from healthy controls and severe COVID-19 patients using scRNA-seq might elucidate the underlying mechanisms. Public scRNA-seq datasets of severe COVID-19 patients could potentially be analyzed to predict potential mechanisms. | In course of this revision whole blood RNA-seq expression profiles were generated from NAPKON patients that were already tested for anti-PGRN/anti-IL-1Ra antibodies and hyperphosphorylated antigen in course of the present study. All related data are presented in a **new Figure S13** and respective results text and discussion as been included in the revised manuscript. Altogether, our analysis was likely affected by a reduced number of samples and high variability. Beyond, to address the raised question, we would rather need to go for RNAseq data of monocytes from previously severely ill and seropositive (or negative) for anti-PGRN/anti-IL-1Ra antibodies. Unfortunately, we do not have such available. However, to better understand underlying genetics which we consider crucial in the study context, we are currently initializing analysis of >3000 data sets from the NAPKON registry in Germany and link those with serological autoantibody status. Yet, this is still in a negotiation phase. | **Results**, page 14 **Discussion**, page 18 | 5 - 11 30 - … | Next to peripheral blood proteomics we also analyzed whole blood gene expression by RNA sequencing. In the NAPKON cohort, positive autoantibody status associated with a pronounced upregulation of the top 50 most significantly differentially expressed genes (**Fig S13 A**, **B**). Gene set enrichment analysis revealed prominent association of severe disease with platelet activation and blood coagulation pathways (**Fig. S13C). However,** when comparing severely ill COVID-19 patients seropositive for autoantibodies with equally sick COVID-19 patients but seronegative for anti-PGRN/anti-IL-1Ra, we observed no significant DEGs (**Fig. S13D)**. When analyzing whole blood transcriptomics among COVID-19 patients enrolled in NAPKON according to anti-IL-1Ra/anti-PGRN status, the observed prominent association of seropositivity for any of the autoantibodies with platelet activation and blood coagulation pathways is likely just reminiscent of the autoantibodies' strong association with severe disease. High disease burden in COVID-19 was already previous associated with a prominent platelet activation and disturbed blood coagulation gene and protein expression signature (59-61). When we compared severely ill COVID-19 patients seropositive for autoantibodies with equally sick COVID-19 patients but seronegative for anti-PGRN/anti-IL-1Ra, we observed no significant differences in gene expression among whole blood cells. Yet, this analysis was likely affected by a reduced number of samples and high variability. To adequately address the question of a causal link of blood cell gene expression, hyperphosphorylation and anti- |

				IL-1Ra/anti-PGRN autoantibodies, an approach involving RNAseq of monocytes from previously severely ill and seropositive (or negative) for anti-PGRN/anti-IL-1Ra antibodies and re-challenged in vitro with inflammatory cytokines to induce hyperphosphorylation may provide more conclusive data. Furthermore, to better understand the role of underlying genetics we are currently in the process of respective data sets from the NAPKON registry alongside serological autoantibody status, but this is yet beyond the scope of the present work.
--	--	--	--	---

IL-1Ra antigenic determinants
recognized in MIS-C and SD^{1,3}

overlap with 2 of 4 anti-IL-1Ra epitopes in IgG4-RD

(Jarrell, ..., Robinson, *JACI*, 2022):

...G⁷³DETRLQLEAVNITDLSEN⁹¹...
...F¹⁰⁰IRSDSGPTTSFESAACPG¹¹⁹...

Rackaityte, ..., DeRisi, 2024, pre-print: ...LQLEAVNITDLSENRK...
Jarrell, ..., Robinson, *JACI*, 2022: ...GDETRLQLEAVNITDLSEN...
our studies¹⁻³: ...GDETRLQLEAVNITDLSEN...

1. Pfeifer*, Thurner B*, Kessel*, ..., Thurner L, *Lancet Rheumatol*, 05/2022
2. Thurner*, Kessel*, ..., Pfeifer**, Klingel**, *NEJM*, 2022
3. Hoffmann, ..., Kessel**, Thurner**, *J Clin Immunol*, 2024

Reply letter Figure 1. Anti-IL-1Ra epitopes identified in independent studies.

Dear Editors, Dear Reviewers,

Once again, thank you very much for critical evaluation of our revised manuscript.

We have addressed all raised points in this point-by-point reply letter.

All changes according to the reviewer's comments are highlighted in **red font**. Additional edits introduced by us to maintain text flow or correct mistakes we have identified outside of reviewer comments are highlighted in **blue font**. In this respect, we have noted a labeling mistake in the previous Figure 4F and G, that pointed out correlations with MCP-3. Yet, this is actually MCP-1, and we corrected this as well as all linked text.

The respective page and line count in this point-by-point reply refers to the track-changes version of the revised manuscript.

Kind regards,

Lorenz Thurner and Christoph Kessel, on behalf of all study authors

Reviewer 2

The authors addressed most comments adequately. However, the answers given to the concern raised around the molecular masses of the proteins detected in the experiments depicted in figure 1 and new figure S7 need further clarification.

- Thank you for your careful evaluation! Happy to do that.

Comment	Authors' reply	New/revised text/item (changes in red font)		
		page/item	line	
The molecular masses of the proteins depicted in figure 1B/D, differ from the molecular masses depicted in the new figure S7D. The latter are in line with immune complexed antigens, the data presented in figure 1B/d do not seem to be in line with this notion. This should be clarified.	Thank you for spotting this! With respect to PGRN, this is also known as granulin–epithelin precursor (GEP) or proepithelin, and is composed of seven homologous granulin (GRN) domains connected by linker regions (i.e. He and Bateman, J Mol Med, 2003). It is proteolytically processed by serine proteases, such as proteinase 3, in a time- and pattern-dependent manner into intermediate fragments and ultimately into individual granulins (GRNs, also termed epithelins) (Kessenbrock et al., JCI, 2008). Of note, phosphorylation of Ser81 also affects PGRN processing (Thurner et al., J Autoimmun, 2015). According to the estimated molecular weights from Figure 1B, only already converted granulin (GRN) is shown. We have clarified this in the figure itself, figure legend as well as results text. In contrast, in the recent gel filtration experiments as well as analysis of artificial immune complexes we detected	Revised Fig. 1 Revised Fig. S7		Figure 1. ... (B) Patients' plasma can contain both processed granulins (GRN, also termed epithelin, approx. 10kDa) as well as progranulin (PGRN, approx. 80kDa), see Figure S8A. ... Figure S7. Characterization of IL-1Ra and PGRN immune complexes. (A-E) FPLC gel filtration of free and antibody-bound IL-1Ra and PGRN. (A) Calibration curve of the Superdex 200 filtration column. (B-E) Affi-Matrix pre-purified plasma samples, either IL-1Ra-Ab pos/PGRN-Ab neg or IL-1Ra-Ab neg/PGRN-Ab pos or seronegative for both anti-IL-1Ra and anti-PGRN antibodies were analyzed by FPLC. Representative chromatograms of IL-1Ra (B) and PGRN immune complexes (C) in seropositive patients' plasma compared to seronegative controls (D, E) are shown. (F, G) Fractions from gel filtration resembling peaks (P1-P3) according to respective chromatograms in B and C were subjected to native gradient PAGE and Western blot in comparison to non-separated seropositive patients' plasma. (G) Similarly, as control fractions from gel filtration of seronegative patients' plasma according to respective chromatograms in D and E were subjected to native gradient PAGE and Western blot in comparison to non-separated

	both PGRN as well as GRN. We have commented on this in the revised results text. With respect to IL-1Ra there was indeed a copy/past error of molecular markers in the previous Figure 1D, which we now corrected. Overall it requires to be stated that our in-house prepared gradient gels may slightly defer between runs (and gels) and may not be 100% robust in determining exact molecular mass. For this reason, we also ran patients' plasma side-by-side with gel filtration separated protein peaks on one and the same gels and analyzed those for either IL-1Ra or PGRN immune complexes (new Figure S7H).			patients' plasma. (H) Side-by-side analysis of gel filtration separated anti-IL-1Ra (left panel) and PGRN immune complexes (right panel) compared to COVID-19 patients' plasma as in Figure 1B and D on native gradient PAGE followed by Western blot.
In figure S7D a band named P4 is depicted. No peak 4 is present in the data presented in figure S7C	Here, we made a mistake when assembling the figure, please excuse! PGRN WB in the previous figure and chromatogram did not match as not from identical samples. The four protein bands in the previous figure S7D included one band for unprocessed PGRN, which we do not observe that frequently. The revised figure S7 now has the correct pairing of FPLC chromatograms and respective native gradient PAGE/WB images.	New Fig. S8		Figure S8. Artificial IL-1Ra and PGRN immune complexes. (A-E) Artificial immune complexes from recombinant antigen and mono- or polyclonal commercial anti-IL-1Ra or anti-PGRN antibody preparations (mAb, pAb) were analyzed by native gradient PAGE and Western blot in comparison to patients' plasma (A). (A) Patients' plasma can contain both processed granulins (GRN, also termed epithelin, approx. 10kDa) as well as progranulin (PGRN, approx. 80kDa). (C) Patient's plasma analyzed for anti-IL-1Ra and anti-PGRN immune complexes on 10% native PAGE followed by Western blot. (D) Native gradient PAGE and Western blot (for IL-1Ra, left panels; for PGRN, right panels) of all components (commercial anti-IL-1Ra and anti-PGRN mAbs, recombinant antigens) used to assemble artificial immune complexes. (E) Native gradient PAGE and Western blot (for IgM, left panels; for IgG, right panels) of all components (commercial anti-IL-1Ra and anti-PGRN mAbs, recombinant antigens) used to assemble artificial immune complexes.
In figure S7D/E, a band around 66 Kd is present. No such band is present in figure 1 of the main manuscript	This accounts for detection of unprocessed PGRN vs. granulin (GRN, processed PGRN) as	Results, page 7	26-30	Native gradient Western-Blots of discovery cohort patients' plasma revealed reduced levels of free (non-immune complexed) granulin (GRN) and IL-1Ra, coinciding with IgG and IgM immune complexed antigen, when plasma samples tested positive for anti-PGRN and anti-IL-1Ra antibodies (Fig. 1B and Fig. 1D)

No negative controls seem to be included in the new dataset depicted in figure S7	discussed above. In the revised manuscript we have attempted to clarify this in both results text as well as Fig 1, Fig S7 and Fig S8. For a side-by-side analysis of protein bands reflecting different immune complexes (as in Figure 1B and D) we also ran selected plasma from these initial analysis next to protein fractions corresponding to respective peaks from gel filtration (new Figure S7H). In the revised Figure S7 we have now also included FPLC and respective native PAGE/WB data of seronegative individuals (Fig S7D, E and G). In the new Figure S8 (focusing on artificial immune complexes) we have included data (Fig. S8D, E) showing WB of native PAGE for the individual components of the artificial ICs.	page 9	26 - ... To further characterize the observed immune complexes in patients' plasma, we performed size-exclusion chromatography in order to separate non-antibody bound and immune-complexed antigens from anti-IL-1Ra/anti-PGRN positive (Fig. S7B, C) as well as seronegative patients (Fig. S7D, E). In case of immune complexed IL-1Ra, we observed three major peaks eluting from seropositive plasma, corresponding to non-immune complexed protein (~20 kDa), IgG- (~177 kDa), and IgM-complexed (~870 kDa) IL-1Ra (Fig. S7B), which we validated by immunoblotting (Fig. S7F, left panel). Similarly, PGRN eluted as unbound protein (~15kDa: granulin, ~65 kDa: progranulin), IgG- (~202 kDa), and IgM-complexed (~903 kDa) PGRN (Fig. S7C and F, right panel). From seronegative plasma proteins eluted as single peaks (Fig. S7D, E) corresponding to IL-1Ra (approx. 18kDa) and unprocessed PGRN (approx., 80kDa; Fig. S7G). Protein fractions as identified by gel filtration corresponded to protein bands as identified in the analysis of COVID 19 patients' plasma in Figures 1B and D (Fig. S7F, H). Furthermore, in order to verify respective sizes of immune complexes detected in patients' plasma, we generated artificial immune complexes using recombinant antigen and mono- or polyclonal commercial IL-1Ra or PGRN-specific immunoglobulins (Fig. S8E-F). These experiments confirmed the banding patterns of Ig-associated species as observed with immune complexes detected in patients' plasma according to native gradient PAGE (Fig S8A, B and D, E) or native PAGE at fixed resolution and subsequent Western blotting (Fig. S8C). It is of note, though, that in seronegative COVID-19 patients included in the initial analysis (Fig. 1B) we only observed a band
--	--	---------------	--

				resembling non-immune complexed granulin (GRN, processed PGRN), whereas in subsequently conducted gel filtration experiments (Fig. S7E, G) and analysis of artificial immune complexes (Fig. 8A) we also observed protein bands resembling both non-immune-complexed GRN and PGRN.
--	--	--	--	--